# GLUT1 overexpression in CAR-T cells induces metabolic reprogramming and enhances potency

Justin A. Guerrero [1,12], Dorota D. Klysz [1,12], Yiyun Chen[1], Meena Malipatlolla[1], Jameel Lone[2], Carley Fowler [1], Lucille Stuani [3], Audre May[1], Malek Bashti [1], Peng Xu[1], Jing Huang[1], Basil Michael[4], Kévin Contrepois[4], Shaurya Dhingra [1], Chris Fisher[1], Katrin J. Svensson [2,5,6], Kara L. Davis [1,3], Maya Kasowski [7,8,9,10], Steven A. Feldman [1], Elena Sotillo [1] ✉ & Crystal L. Mackall [1,3,10,11] ✉

The intensive nutrient requirements needed to sustain T cell activation and proliferation, combined with competition for nutrients within the tumor microenvironment, raise the prospect that glucose availability may limit CAR-T cell function. Here, we seek to test the hypothesis that stable overexpression (OE) of the glucose transporter GLUT1 in primary human CAR-T cells would improve their function and antitumor potency. We observe that GLUT1OE in CAR-T cells increases glucose consumption, glycolysis, glycolytic reserve, and oxidative phosphorylation, and these effects are associated with decreased T cell exhaustion and increased Th$_{17}$ differentiation. GLUT1OE also induces broad metabolic reprogramming associated with increased glutathione-mediated resistance to reactive oxygen species, and increased inosine accumulation. When challenged with tumors, GLUT1OE CAR-T cells secrete more proinflammatory cytokines and show enhanced cytotoxicity in vitro, and demonstrate superior tumor control and persistence in mouse models. Our collective findings support a paradigm wherein glucose availability is rate limiting for effector CAR-T cell function and demonstrate that enhancing glucose availability via GLUT1OE could augment antitumor immune function.

T cells manifest rapid induction of aerobic glycolysis to meet the metabolic needs for proliferation and effector function (Teff) following antigen stimulation[1–5]. This is challenging in the tumor microenvironment (TME), where T cells compete with tumor cells that also rely on aerobic glycolysis[6,7], creating intense competition for glucose[4], compounded by dysregulation of nitrogen metabolism[8,9] and reactive oxygen species (ROS)-mediated T$_{EFF}$ suppression[10,11]. To meet metabolic demand, activated T cells upregulate surface

[1]Center for Cancer Cell Therapy, Stanford Cancer Institute, Stanford University School of Medicine, tanford, CA, USA. [2]Department of Pathology, Stanford University School of Medicine, Stanford, CA, USA. [3]Division of Pediatric Hematology/Oncology/Stem Cell Transplant and Regenerative Medicine, Department of Pediatrics, Stanford University School of Medicine, Stanford, CA, USA. [4]Metabolic Health Center, Stanford University School of Medicine, Stanford, CA, USA. [5]Stanford Diabetes Research Center, Stanford University School of Medicine, Stanford, CA, USA. [6]Stanford Cardiovascular Institute, Stanford University School of Medicine, Stanford, CA, USA. [7]Pediatric Oncology Branch, Center for Cancer Research, National Cancer Institute, Bethesda, MD, USA. [8]Department of Genetics, Stanford University, Stanford, CA, USA. [9]Sean N. Parker Center for Allergy and Asthma Research at Stanford University, Stanford University, Stanford, CA, USA. [10]Division of Bone Marrow Transplant-Cell Therapy, Dept of Medicine, Stanford University School of Medicine, Stanford, CA, USA. [11]Parker Institute for Cancer Immunotherapy, San Francisco, CA, USA. [12]These authors contributed equally: Justin A. Guerrero, Dorota D. Klysz. ✉e-mail: esotillo@stanford.edu; cmackall@stanford.edu

expression of nutrient transporters, which can dictate T cell differentiation[12–16]. Glucose uptake is regulated in large part through expression of the SLC2 (GLUT) family of facilitative glucose transporters[17,18], of which there are 14 members, with GLUT1 (*SLC2a1*) being the most well-studied.

CAR-T cells are genetically engineered to recognize tumor-associated antigen(s) of choice. Like non-engineered T cells, CAR-T cells must sustain the metabolic and energetic needs required for activation, proliferation, differentiation, and killing by balancing glycolysis and oxidative phosphorylation (OXPHOS)[19,20] while competing with tumor cells for nutrients[21–23].

In this study, we find that engineered overexpression of GLUT1 (GLUT1OE) increases glycolytic activity and oxidative phosphorylation in primary human CAR-T cells and induces broad metabolic reprogramming associated with increased inosine accumulation and increased resistance to ROS-mediated immunosuppression. GLUT1OE in CAR-T cells also increases $Th_{17}$ differentiation, diminishes features of exhaustion, and induces greater persistence. When challenged with tumors, GLUT1OE CAR-T cells manifest increased cytokine secretion and superior tumor control in vitro and in vivo compared to control CAR-T cells. These data demonstrate that increased glucose availability in tumor-reactive T cells induces metabolic, transcriptional, and functional reprogramming and provides a new approach to enhance the potency of engineered T cell populations designed for antitumor targeting.

## Results

### GLUT1 overexpression enhances glycolysis and oxidative phosphorylation

We first investigated CAR-T cell dependency on glucose by monitoring the effects of glucose deprivation on expansion of the clinically relevant CD19.28ζ-CAR as well as the high-affinity GD2 targeting HA.28ζ-CAR, which undergoes antigen-independent tonic signaling and manifests hallmark features of T cell exhaustion within 10 days of in vitro culture[24–26]. Both CAR-T cells showed significantly reduced viability and more than 30-fold reduced expansion when grown in media lacking glucose (Fig. 1A), confirming the essential role for glucose as a carbon source for CAR-T cell proliferation in vitro. To determine whether GLUT1OE would increase glucose uptake in CAR-T cells, we co-transduced a CAR-expressing vector and a bicistronic construct containing a selectable marker and GLUT1 separated by a ribosomal skipping site (NGFR-p2a-GLUT1) (Fig. 1B). Compared to control cells, CAR-T cells overexpressing GLUT1 demonstrated increased intracellular glucose uptake as measured using the fluorescent glucose analog 2-NBDG and deoxy-D-[1,2-3H (N)]-glucose at baseline. (Fig. 1C, D). Antigen-mediated stimulation of CD19.28ζ CAR-T cells increased the amount of glucose uptake, which was further enhanced by GLUT1OE. HA.28ζ CAR-T cells showed higher GLUT1 expression and higher 2-NBDG uptake at baseline compared to CD19.28ζ CAR-T cells, and antigen stimulation of HA.28ζ CAR-T cells induced a lesser effect on total glucose uptake, consistent with increased energetic needs at baseline in HA.28ζ CAR-T cells due to tonic signaling.

We next utilized Seahorse analysis to measure glycolytic capacity in CD19.28ζ and HA.28ζ CARs ± GLUT1OE (CD19.28ζ-GLUT1 and HA.28ζ-GLUT1). Basal extracellular acidification rate (ECAR) was unchanged with GLUT1OE in CD19.28ζ, in contrast to the tonically signaling HA.28ζ CAR-T cells, which demonstrated higher basal ECAR that was further enhanced by GLUT1OE (Fig. 1E). Both CD19.28ζ and HA.28ζ CAR T cells manifested increased glycolytic reserve with GLUT1OE (Fig. 1F), whereas untransduced "Mock" T cells with GLUT1OE did not exhibit any of the aforementioned changes when compared to control, illustrating a substantial impact of CAR expression on GLUT1OE mediated metabolic programming (Supplementary Fig. 1A, B). Activation-induced T cell effector programming requires

glycolysis which increases glucose demand[27], thus we measured glycolytic flux ± GLUT1OE following antigen-mediated CAR activation using anti-idiotype antibodies. GLUT1OE-CD19.28ζ and -HA.28ζ CAR-T cells manifested a greater deltaECAR ($ECAR_{max}$ during activation minus $ECAR_{max}$ at baseline) compared to controls (Fig. 1G), although the deltaECAR in HA.28ζ-GLUT1 cells remained below that observed in CD19.28ζ ± GLUT1OE.

To further investigate changes in the metabolic state induced by GLUT1OE, we used Seahorse analysis to quantify mitochondrial respiration. GLUT1OE increased basal and maximum oxygen consumption rate (OCR) and spare respiratory capacity (SRC) in CD19.28ζ and HA.28ζ CAR-T cells compared to controls (Fig. 2A), whereas GLUT1OE in Mock T cells reduced basal OCR (Supplementary Fig. 1C). Although we observed no change in mitochondrial mass, GLUT1OE induced significantly higher mitochondrial potential in HA.28ζ, but not CD19.28ζ (Fig. 2B, C). Together, these data demonstrate that glucose availability is rate-limiting in CAR-T cells following antigen-driven activation since GLUT1OE enhances glycolysis and mitochondrial respiration in this setting. They further demonstrate that the effects of GLUT1OE are more profound in CD19.28ζ CAR-T cells compared to chronically activated HA.28ζ CAR-T cells, which we attribute to increased GLUT1 expression at baseline in response to chronic rate limiting glucose availability in tonic signaling CAR-T cells.

### GLUT1 overexpression decreases the transcriptional program associated with T cell exhaustion and increases $Th_{17}$ differentiation

To characterize how and to what extent GLUT1OE-induced augmentation of glycolysis and mitochondrial respiration associates with changes in gene expression, we conducted bulk RNA sequencing (RNAseq) in CD19.28ζ and HA.28ζ CAR-T cells ± GLUT1OE at baseline and after 4 or 14 h of CAR stimulation. As expected, activation-induced dominant effects on gene transcription, as revealed by unbiased PCA clustering, which showed three distinct populations representing baseline and the two post-activation time points (Fig. 3A). As we have previously shown[25,26], the transcriptional programs of CD19.28ζ and HA.28ζ CAR-T cells at baseline are very different, resulting in different *starting points* upon which the transcriptional reprogramming of GLUT1OE occurred and distinct patterns for each CAR. Nonetheless, focusing on the unstimulated state, we identified a common set of ~800 genes regulated by GLUT1OE in both CAR T cells (Fig. 3B). GSEA analysis of this gene set showed down-regulation of the NK-like exhaustion signature (including *GNLY* and *TNFRSF9*) in both CAR T cells upon GLUT1OE (Fig. 3C)[28]. Consistent with the Seahorse analyses, CD19.28ζ-GLUT1 T cells at rest and following activation demonstrated increased expression of genes related to glycolysis and OXPHOS (Fig. 3D) and a shift from a naïve T cell transcriptional signature towards one associated with memory and effector cells (Fig. 3E). We also observed induction of genes associated with numerous metabolic pathways, including arginine, glutamate, glutathione and fatty acid metabolism among others. Most of these metabolic changes were more evident at baseline and 4 h after activation (Fig. 3E, F).

We next sought to determine whether these transcriptional effects were associated with modulation of T cell differentiation. We found that CD19.28ζ-GLUT1 CAR-T cells showed upregulation of $Th_{17}$-associated cytokines *IL17F and IL22* even before activation, and increased transcription of *IL17A* and genes associated with $Th_{17}$ differentiation after antigen exposure (Fig. 4A, B). GLUT1OE also increased protein expression of CCR4+ and CCR6+ in both CD19.28ζ and HA.28ζ GLUT1OE CAR-T cells at baseline and after idiotype stimulation (Fig. 4C, D) and CD19.28ζ-GLUT1OE T cells secreted higher levels of IL-17A and IL-17F after stimulation with Nalm6 (Fig. 4E–G).

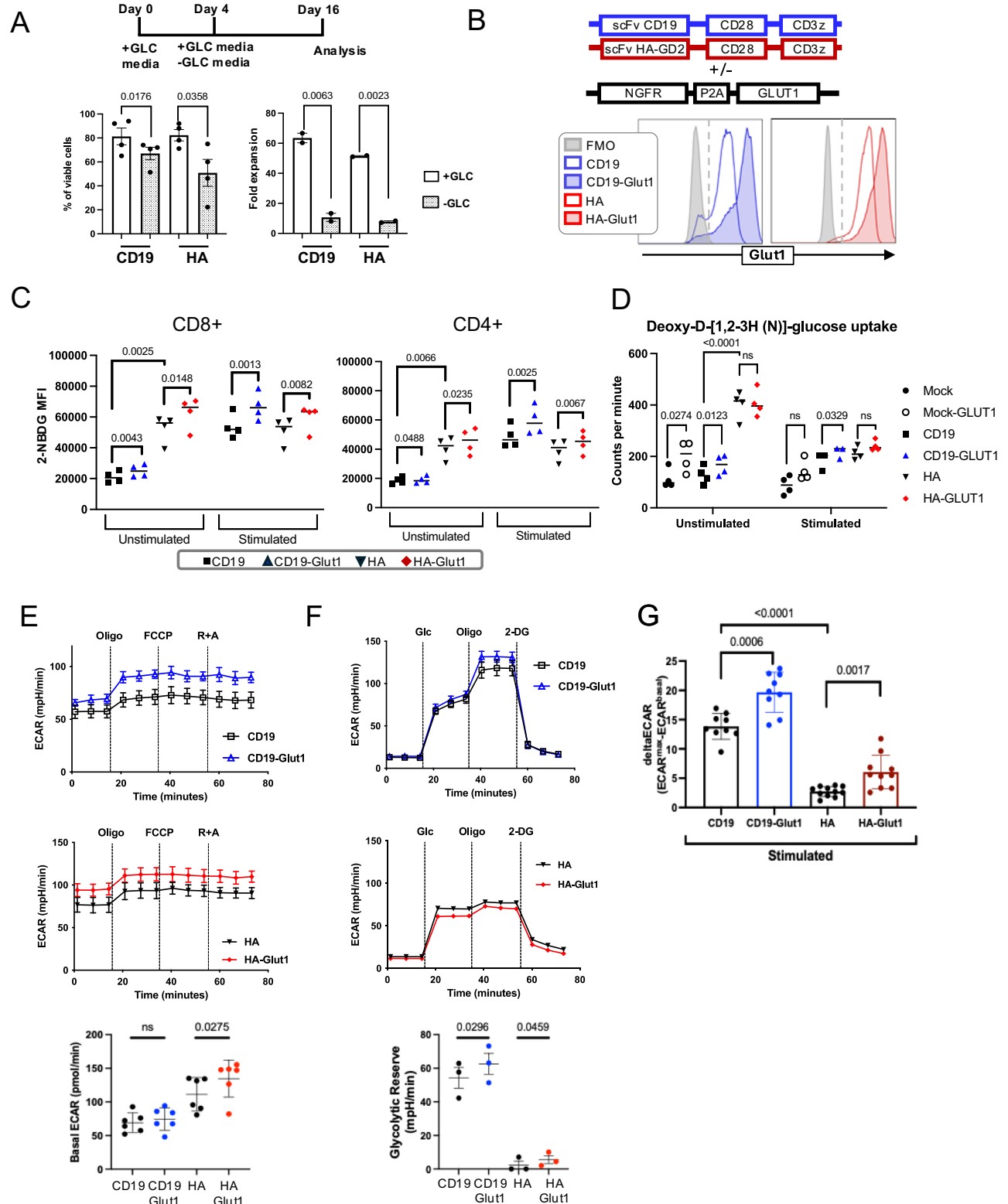

Interestingly, although HA.28ζ-GLUT1 cells showed increased surface expression of CCR4 and CCR6, they secreted increased levels of Th₁ and Th₂-associated cytokines and IL-22 compared to control cells following antigen challenge (Fig. 4G, H). Collectively, the data demonstrate that GLUT1OE reduces the transcriptional signature associated with exhaustion, induces Th₁₇ differentiation, and promotes metabolic reprogramming associated with increased glycolysis and increased oxidative phosphorylation.

## GLUT1 overexpression enhances metabolic pathways that favor resistance to REDOX suppression

GLUT1OE CAR-T cells manifested increased GLUT3 (*SLC2A3*) and ATP Synthase (*ATP5F1B*) expression in CD19.28ζ-GLUT1 post-stimulation (Supplementary Fig. 2A, B), as well as upregulation of glutathione synthetase (*GSS*) and cystathionase (*CTH*) transcripts, two enzymes involved in production of the antioxidant glutathione (GSH), which serves as the cells' principal defense against ROS imbalance[29]

**Fig. 1 | GLUT1 overexpression enhances glycolysis. A** (TOP) Schematic of experimental design: CAR-T cells were activated in the presence of glucose and then cultured in media ± glucose starting on day 4. Glucose concentration is 11 mM for all experiments unless otherwise noted. (BOTTOM) Viability and fold expansion on day 16. Pooled data of $n = 2$–4 donors. $P$ values determined by paired two-tailed $t$-tests. Error bars represent SD. **B** (TOP) Schema of retroviral vectors expressing CAR and NGFR-P2A-GLUT1. NGFR is used as a selectable membrane marker of GLUT1-OE cells. (BOTTOM) Representative flow cytometry histogram of GLUT1 surface expression using a GLUT1-specific H2RBD-GFP ligand, in control CD19 and HA CAR T cells ± NGFR-GLUT1 vector (CD19-GLUT1, HA-GLUT1). Analysis of GLUT1OE CAR T cells performed by gating on CAR+/NGFR+ populations (percentage of double-positive >80% for each experiment unless otherwise noted). FMO Fluorescence minus one control. **C** 2-NBDG median fluorescence intensity of (LEFT) CD8+ and (RIGHT) CD4+ control HA and CD19 CAR T cells and double positive gated NGFR+CAR+ cotransduced CD19-GLUT1, HA-GLUT1 ± 24H idiotype stimulation (1 μg/mL). Pooled data of $n = 4$ donors. $P$ values determined by paired

two-tailed $t$-tests. **D** Pooled data reflecting glucose uptake in CAR-T cells ± 24 h stimulation with idiotype using deoxy-D-[1,2-3H(N)]-glucose. Data from $n = 4$ donors. $P$ values determined by paired two-tailed $t$-tests. **E** (TOP) Representative extracellular acidification rate (ECAR) measured using Seahorse for CD19 ± GLUT1OE on day 14 from one donor. (MIDDLE) ECAR for HA ± GLUT1OE. (BOTTOM) Pooled data for basal ECAR. Data are representative of three independent experiments with $n = 6$ donors. $P$ values determined by paired two-tailed $t$-tests. Error bars represent SD. **F** (TOP) Representative ECAR (Glycolytic Stress Test) measured using Seahorse for CD19 ± GLUT1OE on day 12 from one donor. (MIDDLE) ECAR for HA ± GLUT1OE. (BOTTOM) Pooled glycolytic reserve data. Data are representative of one experiment with $n = 3$ donors. $P$ values determined by paired two-tailed $t$-tests. Error bars represent SD. **G** ECAR measured at the baseline and 3 minutes after stimulation with 10 μg/ml of anti-idiotype crosslinked with 10 μg/mL of anti-mFAB on day 16. $P$ values determined by unpaired two-tailed $t$-tests. Error bars represent SD.

(Fig. 5A, B). Consistent with this, CD19.28ζ-GLUT1 CAR-T cells exhibited increased intracellular GSH, as measured by thiol staining, and mass spectrometry analysis showed that both CD19.28ζ-GLUT1 and HA.28ζ-GLUT1 contained less of the oxidized form of GSH, cysteineglutathione disulfide (GSSG) (Fig. 5C, D). Intracellular staining also demonstrated less mitochondrial ROS in CD19.28ζ-GLUT1 cells compared to control post stimulation, and GLUT1OE had the same effect for HA.28ζ at baseline (Fig. 5E). These findings were validated using CyTOF single cell proteomic profiling[30], which confirmed that GLUT1OE CD8+ and CD4+ CAR-T cells expressed increased levels of GSS, GLUT1 and GLUT3, and increased expression of the OXPHOS-associated ATP5F1B (a subunit of ATP Synthase) (Supplementary Fig. 2C). Following stimulation, CD19.28ζ-GLUT1 and HA.28ζ-GLUT1 CAR-T cells continued to express higher levels of GLUT1, GLUT3, and CD62L compared to controls, while ATP5 and PPP-associated G6PD were unchanged (Supplementary Fig. 2D).

Glutaminolysis metabolizes glutamine to glutamate, a metabolite essential to GSH formation in the presence of cysteine. To assess whether increased glutaminolysis might contribute to increased ROS in GLUT1OE CAR-T cells, we analyzed our RNAseq dataset and observed that GLUT1OE CD19.28ζ CAR-T cells expressed higher levels of genes involved in glutaminolysis, including GOT2, GLUD1, and GPT2, both at baseline and after antigen stimulation (Fig. 5F). Further evidence in support of a model wherein GLUT1OE increases glutaminolysis emerged from our mass spectrometry data, which showed less glutamine in CD19.28ζ-GLUT1 CAR-T cells. Together, these data are consistent with a model wherein GLUT1OE increases glutaminolysis and antioxidant production (Supplementary Fig. 5G).

Transcriptomic, metabolomic, and proteomic findings suggested that GLUT1OE may endow resistance to ROS accumulation. To test this hypothesis, we subjected CD19.28ζ CAR ± GLUT1OE to $H_2O_2$ prior to antigen challenge with Nalm6 cells. As a control, we treated cells with catalase, which rapidly mediates the conversion of $H_2O_2$ into $O_2$ and $H_2O$, immediately before exposure to ROS. Both CD8+ and CD4+ CD19.28ζ-GLUT1 cells were more resistant to $H_2O_2$ suppression as measured by increased IL-2 secretion, and this effect was abrogated in the presence of L-buthionine-S,R-sulfoximine (BSO), which diminishes GSH levels by inhibiting the catalytic subunit of glutamate–cysteine ligase (GCL) and GSH biosynthesis (Fig. 5H–J, Supplementary Fig. 3A, B). To determine whether the enhanced cytokine secretion observed in this assay was dependent on glucose supporting pentose phosphate pathway (PPP) activity, we challenged CD19.28ζ ± GLUT1OE against Nalm6 in the presence of 6-aminonicotinamide (6-AN), an inhibitor of 6PGD. CD19.28ζ-GLUT1 continued to secrete more cytokines as compared to control, with no meaningful suppression in the presence of 6-AN, demonstrating that the PPP was not required for these findings (Fig. 5K). Together, these results demonstrate that GLUT1OE induces antioxidant-

promoting pathways that endow resistance to ROS-induced suppression, which are predicted to enhance antitumor potency.

## GLUT1 overexpression alters arginine and inosine metabolism

We next directly examined alterations in the metabolome induced by GLUT1OE using global mass spectrometry. Consistent with RNAseq and single cell proteomic analyses, we identified numerous metabolites and pathways that were differentially abundant in CD19.28ζ-GLUT1 and HA.28ζ-GLUT1 CAR-T cells compared to controls (Fig. 6A, Supplementary Fig. 4A). Glycine and serine metabolism were among the most enriched pathways, both of which provide crucial substrates for GSH production and REDOX homeostasis, consistent with the functional data demonstrating enhanced resistance to REDOX stress in cells with GLUT1OE[31–33]. We also found that metabolites involved in the urea cycle were highly enriched, including homoarginine, dimethylarginine, citrulline, and acetylornithine, although arginine was reduced in GLUT1OE cells compared to controls (Fig. 6B, C). The significant increase in homoarginine suggested that the decreased arginine levels in GLUT1OE cells were likely due to its conversion to homoarginine, and not due to decreased biosynthesis from citrulline, since two upstream rate limiting enzymes argininosuccinate synthase 1 (ASS1) and argininosuccinate lyase (ASL) were increased by GLUT1OE in CD8+ GLUT1OE CARs and CD19.28ζ-GLUT1 respectively, nor from decreased arginine demand, as transcription of the main arginine transporter SLC7A1 was significantly upregulated (Fig. 6D, E, Supplementary Fig. 4B, C).

To assess the downstream effects of the observed changes in urea cycle activity, we analyzed the effect of GLUT1OE on the activity of MTOR, a master regulator of T cell proliferation, survival, and metabolism that can be regulated by glycolytic metabolism and nutrient availability[34]. As predicted by RNAseq and metabolomic analysis, flow cytometry confirmed increased phosphorylated ribosomal subunit S6 (pS6) in HA.28ζ-GLUT1 cells up to 24 h after CAR activation (Fig. 6F). To further assess the downstream consequences of increased glucose uptake in CD19.28ζ-GLUT1 CAR-T cells, we utilized [U13C] isotopically labeled glucose for carbon tracing (Fig. 7). GLUT1OE promoted lactic acid formation, and glutamate derived from TCA activity. The biosynthesis of inosine, indirectly formed through the PPP, was also found in significantly greater abundance in CD19.28ζ-GLUT1, and significantly so after stimulation. Our data demonstrate GLUT1OE induces extensive alterations in glucose-derived metabolism spanning carbon cycling though the PPP, TCA and urea cycles associated with broad metabolic reprogramming in GLUT1OE CAR-T cells, rather than narrow effects on one specific metabolic enzyme or pathway.

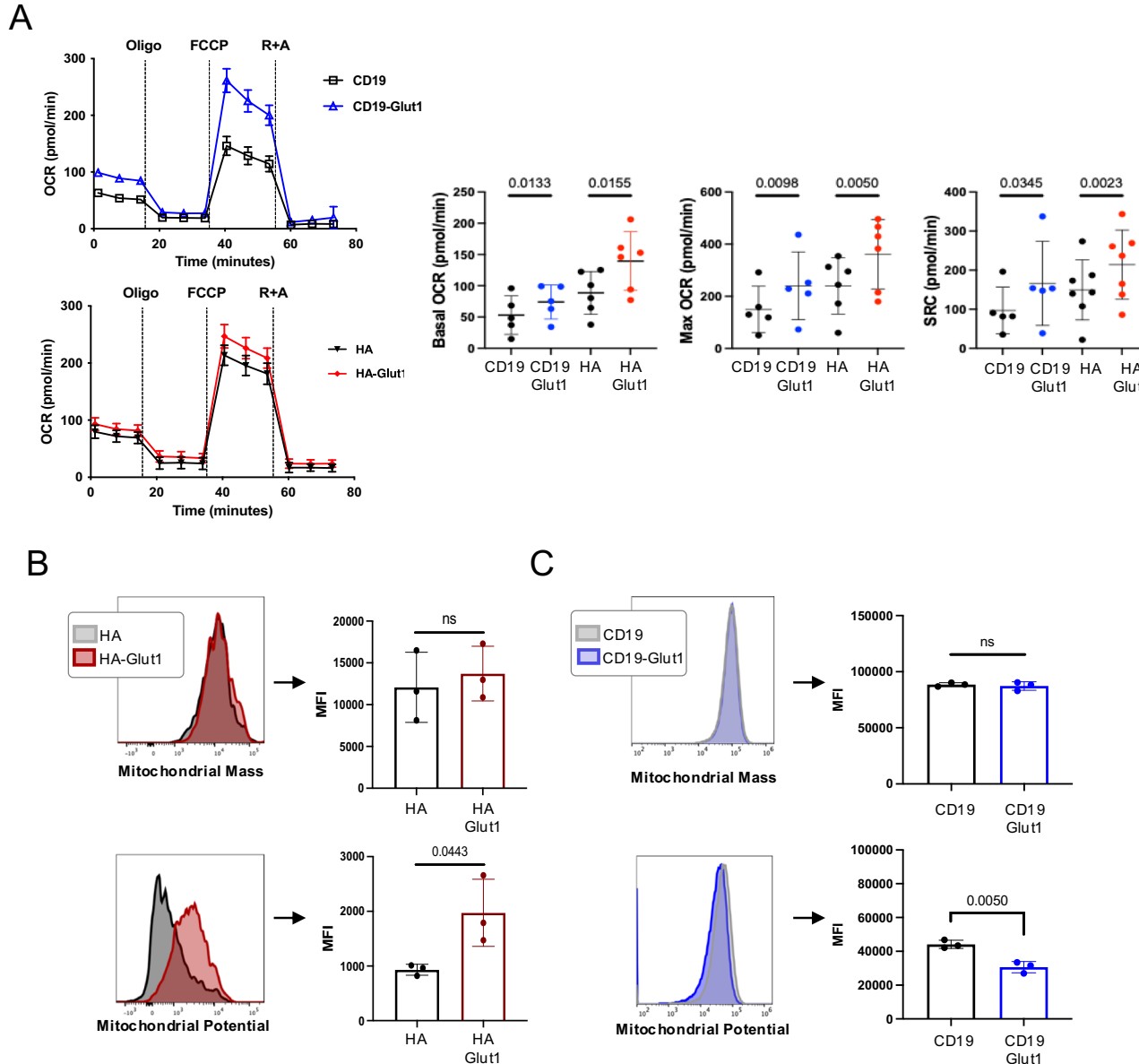

**Fig. 2 | GLUT1 overexpression enhances oxidative phosphorylation. A** (LEFT) Representative OCR (Oxygen Consumption Rate) data measured using Seahorse for CD19 ± GLUT1OE or for HA ± GLUT1OE on day 12. (RIGHT) Pool data for Basal and Maximum OCR and SRC (Spare Respiratory Capacity) from three independent experiments with *n* = 6 donors. *P* values determined by paired two-tailed t-tests. Error bars represent SD. **B** (TOP) Mitochondrial mass and (BOTTOM) potential detected using Mitotracker Green and Deep Red, respectively in HA ± GLUT1OE

CAR-T cells on day 9 with representative histograms shown. Data from *n* = 3 donors. *P* values determined by paired two-tailed t-tests. Error bars represent SD. **C** (TOP) Mitochondrial mass and (BOTTOM) potential detected using Mitotracker Green and Deep Red, respectively in CD19 ± GLUT1OE CAR-T cells on day 9 with representative histograms shown. Data from *n* = 3 donors. *P* values determined by paired two-tailed t-tests. Error bars represent SD.

## GLUT1 overexpression increases potency in response to tumor challenge

We next sought to determine whether the transcriptional and metabolic reprogramming induced by GLUT1OE would endow them with enhanced antitumor immunity, and/or predispose CAR-T cells to exhaustion, as some studies suggested based on glucose uptake restriction[35]. Analysis of the expression of canonical exhaustion markers at baseline by flow cytometry showed no differences or lower expression upon GLUT1OE in both CD4+ and CD8+ CAR-T cells (Fig. 8A, Supplementary Fig. 6A). Next, we challenged CD19.28ζ-GLUT1, HA.28ζ-GLUT1, and their respective controls with CD19+ or GD2+ Nalm6 leukemia, respectively. Both GLUT1OE CARs demonstrated marked increases in tumor induced IL-2 and IFNγ secretion (Fig. 8B). Similar results were seen following challenge with the

CD19+GD2+ osteosarcoma line 143b (Fig. 8C). Accordingly, a greater proportion of CD19.28ζ-GLUT1 and HA.28ζ-GLUT1 produced IL-2 and TNFα compared to controls, as measured via intracellular staining, following challenge with Nalm6-GD2 at differing E:T ratios (Fig. 8D).

We next assessed if GLUT1OE would accelerate the onset of exhaustion upon tumor rechallenge. CAR-T cells were sequentially co-cultured at 1:2 E:T ratio with Nalm6-GFP ± GD2, tumor growth was assayed using Incucyte, and CD39 and PD-1 were measured by flow cytometry upon tumor clearance (Fig. 8E). Flow cytometry showed that both CD19.28ζ and HA.28ζ GLUT1OE CARs expressed less CD39 and PD-1 compared to control cells across time points, especially in the CD4+ compartment (Fig. 8F, Supplementary Fig. 6B). At ~200 h post initial culture (after 4 stimulations for CD19 cells, and 3 for HA),

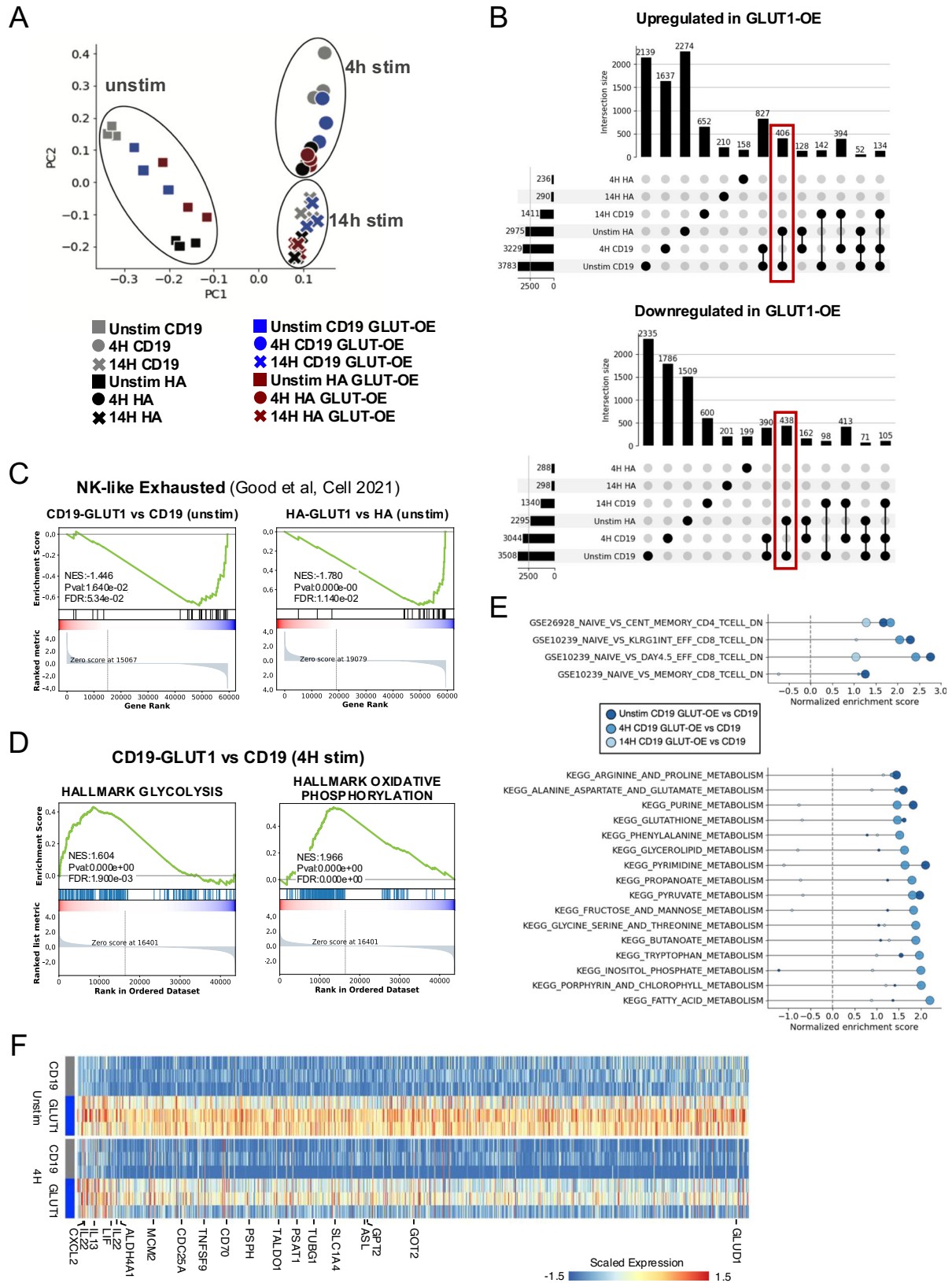

CD19.28ζ-GLUT1 manifested significantly higher proportions of CD8[+] and CD4[+] central memory cells and HA.28ζ-GLUT1 CAR-T cells contained more CD4[+] effector and central memory cells when compared to control (Fig. 8G, Supplementary Fig. 6C). HA.28ζ-GLUT1 CAR-T cells also controlled tumor more efficiently than control cells after the first challenge but the enhanced potency was lost after three total challenges. Based on these findings, we analyzed other stemness-associated markers in CAR-T cells at baseline and observed that CD8[+] HA.28ζ-GLUT1 exhibited significantly higher mean expression of TCF1, a transcription factor associated with the formation of

**Fig. 3 | GLUT1 overexpression induces transcriptional reprogramming.**
**A** Unbiased principal component analysis of bulk RNA derived from day 16 CD19 and HA ± GLUT1OE ± 1 µg / mL anti-idiotype stimulation collected at two different time points. Cotransduced cells were magnetically enriched for greater than 95% double positive prior to experiment. Pooled data from two experiments (stimulated and unstimulated) with total $n = 6$ donors. **B** UpSets plots showing intersection of genes differentially upregulated (TOP) or downregulated (BOTTOM) upon GLUT1OE in HA-CAR and CD19-CAR T cells unstimulated or at 4 h and 14 h post stimulation. Red boxes highlight shared changes between CD19-CAR and HA-CAR T cells as a consequence of GLUT1OE at baseline. RNAseq data from $n = 6$ donors on day 16. **C** GSEA analysis of the NK-like exhaustion signature in unstimulated (LEFT)

CD19 and (RIGHT) HA-CAR-T cells, comparing GLUT1OE versus control. **D** GSEA analysis of (LEFT) glycolysis and (RIGHT) OXPHOS for CD19 ± GLUT1OE after 4 h of anti-idiotype stimulation (1 µg/mL). **E** (TOP) GSEA analysis of RNA-seq comparing CD19 GLUT1OE vs CD19 showing enrichment of memory and effector T cell signatures over naïve in CD4 and CD8 at every timepoint analyzed (unstimulated, 4 h or 14 h post-stimulation). (BOTTOM) Similar GSEA analysis using as reference KEGG pathways dataset showing wide metabolic reprogramming. The size of the dots correlates with −log10(P-value) by GSEA analysis, with the smallest dots representing non-significant pathways. **F** Heatmap representing differentially expressed genes with annotations for those significantly upregulated in CD19-GLUT1 ± 4 h of anti-idiotype stimulation.

immunological memory[36] that is typically reduced in terminally exhausted cells (Fig. 8H, Supplementary Fig. 6C). Furthermore, TCF1 was increased in GLUT1OE T cells in the absence of any CAR receptor. We also found significantly elevated expression of memory-associated CD62L in CD8$^+$ and CD4$^+$ HA.28ζ-GLUT1 cells at levels comparable to CD19.28ζ (Fig. 8I, Supplementary Fig. 6D). Together our data provide no evidence that GLUT1OE predisposes T cells to exhaustion, but rather demonstrate that GLUT1OE is associated with diminished expression of exhaustion programs, decreased terminal differentiation, and greater functionality with repetitive stimulation.

Given the evidence for enhanced potency of CD19.28ζ-GLUT1 and HA.28ζ-GLUT1 CAR-T cells, we next tested the effect of GLUT1OE on GPC2.28ζ CAR-T cells which we previously showed were sensitive to low levels of antigen density[37]. Using a bicistronic construct (Fig. 9A) as described above, GPC2.CD28ζ+/− GLUT1 CAR-T cells were co-cultured with neuroblastoma cell lines expressing ~34,000 (NGP-GPC2) or 6800 (SMS-SAN) molecules of GPC2 on the surface. After 24 h, GPC2.28ζ-GLUT1 cells secreted significantly more IL17A against SMS-SAN and more IL-2 and IFNγ against NGP-GPC2 (Fig. 9B, C). In cytotoxicity assays, GPC2.28ζ-GLUT1 completely controlled growth of antigen-low SMS-SAN cells in vitro at a 1:5 E:T ratio, while GPC2.28ζ did not (Fig. 9D). GPC2.28ζ-GLUT1 also improved tumor clearance of NGP-GPC2 cells across multiple E:Ts (Fig. 9E). Together, these data provide convincing evidence that GLUT1OE manifest enhanced antitumor potency associated with increased cytokine secretion and enhanced cytotoxic activity against solid tumors in vitro.

### GLUT1 overexpression enhances CAR-T cell tumor clearance in vivo

We next tested the effect of GLUT1OE in a mouse stress test model, wherein suboptimal doses of CD19.28ζ ± GLUT1OE ($0.350 \times 10^6$/mouse) were infused in NSG mice bearing Nalm6 leukemia (Supplementary Figs. 6A, B). Although neither CAR controlled tumor outgrowth, CD19.28ζ-GLUT1 CAR-T cells mediated a significantly greater delay in tumor growth compared to CD19.28ζ CAR or MOCK +/−GLUT1 T cells (Fig. 10A), and analysis of total splenocytes at endpoint revealed a significantly higher proportion of CAR-T cells, and lower levels of residual Nalm6, in mice treated with CD19.28ζ-GLUT1 (Fig. 10B, C). We next tested whether GLUT1OE enhanced antitumor activity of HA.28ζ in vivo by engrafting Nalm6-GD2 cells into NSG mice (Supplementary Fig. 6C, D). HA.28ζ-GLUT1 demonstrated long-term tumor control (Fig. 10D) and persistence in the peripheral blood on day 25 and day 40 (Fig. 10E). On day 52 surviving mice from the HA.28ζ-GLUT1 CAR-T group were re-challenged with Nalm6-GD2 and continued to exhibit complete anti-tumor immunity for 8 days post re-challenge. On day 60 the same mice were split into two groups for a second rechallenge with antigen positive Nalm6-GD2 ($n = 2$) or antigen negative Nalm6 ($n = 3$). Once again, we observed antigen specific protection, consistent with immunologic memory (Fig. 10F). Lastly, we tested the efficacy of GPC2-28ζ ± GLUT1OE CAR-T cells against the antigen low neuroblastoma cells SMS-SAN engrafted on the kidney capsule of NSG mice

(Supplementary Fig. 6E, F). In this solid tumor model, we observed complete control of outgrowth independent of GLUT1OE (Fig. 10G), however, blood analysis on days 18 and 34 post-tumor engraftment showed higher levels of circulating human T cells, and T$_{SCM}$ populations in mice treated with GPC2-28ζ-GLUT1 as compared to mice administered control CAR-T cells (Fig. 10H). Collectively, these data demonstrate that GLUT1OE enhances CAR T cell potency and persistence in vivo.

## Discussion

The remarkable progress achieved with the use of adoptive cell therapy for B cell and plasma cell malignancies is driving new approaches to leverage cell engineering to enhance T cell potency. One such approach involves harnessing our knowledge of metabolism in T cells to better equip them to sustain the demands of activation, persistence, high tumor burdens and suppressive TMEs. Activated T cells undergo significant metabolic shifts that depend on glucose catabolism[38–41]. Decreased glucose availability can decrease T$_{EFF}$ proliferation, suppress IFNγ secretion[19,42] and mTORC1 activity following stimulation leading to weakened immune responses[43]. In this study, we sought to engineer CAR-T cells to be more competitive in TMEs characterized by limiting availability of glucose. GLUT1 is a major regulator of activation-induced glycolysis and transgenic overexpression in mice increases IL-2 secretion in response to TCR stimuli and increases proliferation of human T cells[3,16,44–46]. Based upon these data, we overexpressed GLUT1 to assess whether glucose availability is rate limiting for CAR-T cell potency and whether this maneuver could enhance CAR-T cell potency.

GLUT1 overexpression enhanced both glycolysis and oxidative phosphorylation, most notably following T cell activation, illustrating the degree to which glucose availability tunes the magnitude of immune responses in CAR T cells. Alterations in metabolism can also skew T cell differentiation lineages and phenotype[47]. Consistent with this, we observed that GLUT1OE drove greater differentiation into Th$_{17}$ cells. Although novel, this finding aligns with previous evidence that aerobic glycolysis is indispensable for Th$_{17}$ differentiation[48,49], and GLUT1 expression is elevated in Th$_{17}$ T cells[3].

Enhanced glycolysis induced by GLUT1OE could predispose to T cell exhaustion[50], especially for CAR-T cells which undergo tonic signaling. We sought to address this here by evaluating whether GLUT1OE led to features typically associated with exhaustion, and we found no evidence for such. Indeed, our evidence demonstrates that GLUT1OE CAR-T cells show decreased expression of transcriptional profiles associated with exhaustion, delayed exhaustion marker expression, increased memory differentiation after repeated tumor challenge and increased persistence in vivo. Furthermore, tumor rechallenge of animals following control by GLUT1OE T cells was associated with sustained antitumor efficacy providing evidence for long term functionality in murine models. These results align with recent studies showing that GLUT1OE is associated with increased memory formation in CAR T cells and enhanced antitumor potency in mouse models of acute lymphoblastic leukemia (ALL), renal cell carcinoma (RCC), and glioblastoma (GBM)[51]. Similarly, enforced GLUT3 expression in OT1

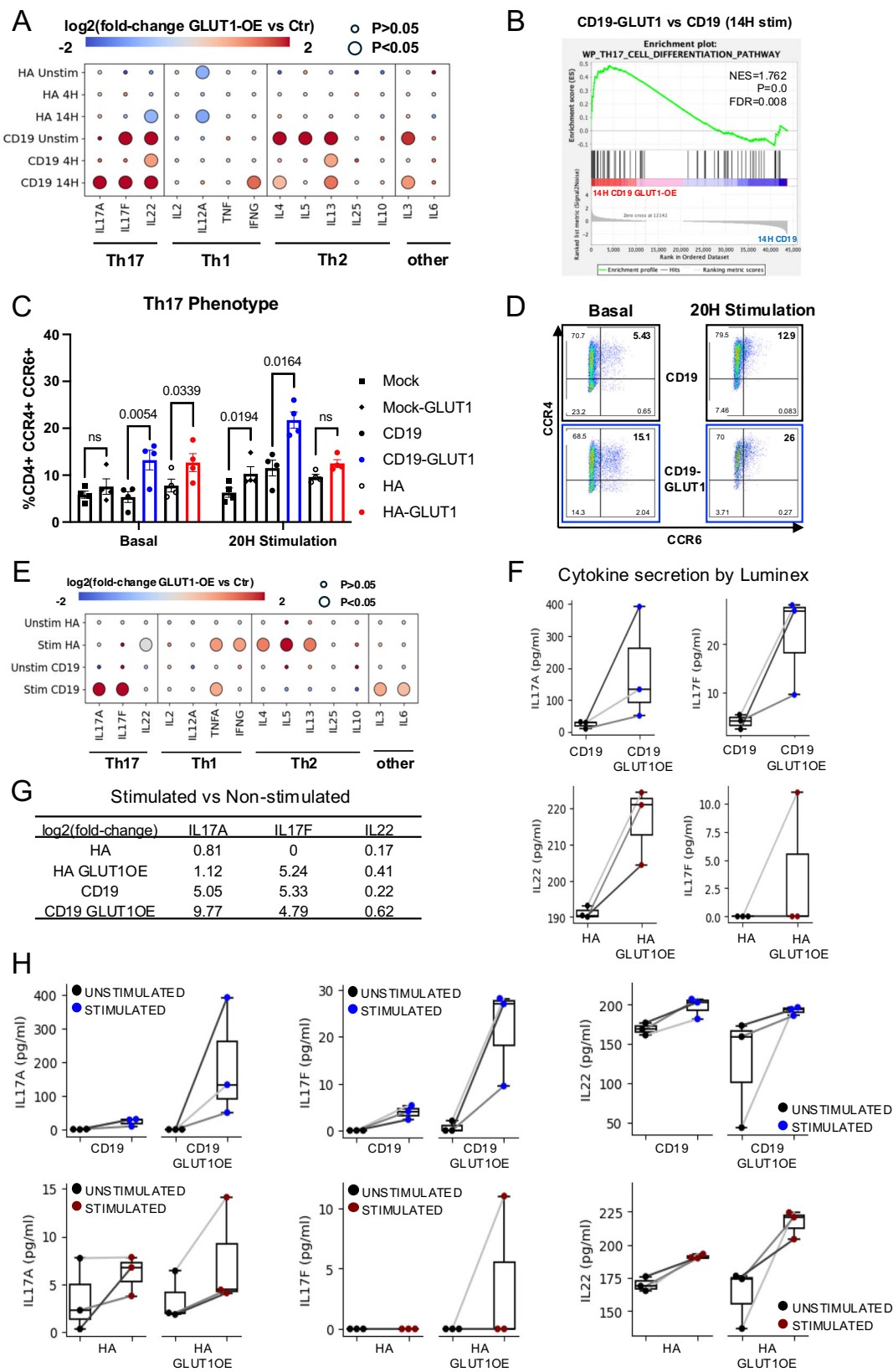

T cells promoted a sustained memory phenotype after repeated antigen stimulation in vitro and enhanced protection after tumor rechallenge in vivo[52].

Beyond the expected enhancement in glycolysis and oxidative phosphorylation induced by GLUT1OE CAR-T cells, we also observed global transcriptional and metabolic reprogramming including upregulation of *ALDH4A1, GOT1, CTH, RETSAT,* and *VDAC3,* transcripts implicated in REDOX biology[53–57] and resistance to REDOX stress, results that are predicted to enhance antitumor potency, as previously shown by engineering catalase overexpressing CAR-T cells[58]. The metabolic pathways responsible for enhanced resistance to REDOX stress with GLUT1OE remain unclear. Reversible cycling of the

**Fig. 4 | GLUT1 overexpression induces Th$_{17}$ differentiation. A** Bubble plot highlighting the changes in cytokine expression for CD19 and HA ± GLUT1OE CAR-T cells, ± idiotype stimulation. The color of the bubble represents the fold-change of expression, while the size represents statistical significance assessed by Wilcox's rank-sum test. **B** GSEA analysis of Th$_{17}$ signatures in CD19-CAR T cells with GLUT1OE versus control, at 14 h post-stimulation. **C** Pooled data of CD4$^+$ CCR4$^+$ CCR6$^+$ Mock and CAR-T cells on day 17 (baseline) and after 20 h idiotype stimulation. Data from $n = 4$ donors. $P$ values determined by paired two-tailed $t$-tests. Error bars represent SEM. **D** Representative flow cytometry of CD4$^+$ gated CD19 ± GLUT1OE CAR-T cells showing Th$_{17}$ phenotype. **E** Bubble plot highlighting the changes in cytokine secretion for CD19 and HA ± GLUT1OE CAR-T cells, ±Nalm6 challenge as captured by Luminex. The color of the bubble represents the fold-change of expression, while the size represents statistical significance assessed by Wilcox's rank-sum test. **F** Boxplots of Th$_{17}$-related cytokines in CD19-CAR and HA-CAR T cells ± GLUT1OE after 24 h stimulation with Nalm6 as captured using Luminex. Data points from the matched donors are connected with lines. **G** The log2(fold-change) in IL17A, IL17F, and IL22 expression in stimulated versus unstimulated state in each CAR-T cell. **H** Boxplots of Th$_{17}$-related cytokines in stimulated versus unstimulated CD19-CAR and HA-CAR-T cells, with and without GLUT1-OE. Data points from the matched donors are connected with lines.

antioxidant GSH to GSSG is the primary mechanism of REDOX balance of superoxide-derived hydrogen peroxide (H$_2$O$_2$). Previous data has demonstrated that glucose consumption cycling via the pentose phosphate pathway (PPP) can alter the GSH:GSSG ratio in favor of antitumor activity[59]. However, we found that inhibition of PPP activity using 6-AN did not alter CD19.28ζ-GLUT1's advantage in cytokine secretion when compared to control. Glutamate, produced through glutaminolysis and/or TCA, can also contribute to biosynthesis of GSH[60,61]. Although we were not able to definitively implicate glutaminolysis in resistance to REDOX stress, we did observe evidence for increased glutaminolysis in GLUT1OE CAR T cells, suggesting that this pathway could be involved in increasing GSH levels in CD19.28ζ-GLUT1 CAR-T cells.

In summary, the field is developing an army of next-generation CARs utilizing rapidly developing technologies aimed at increasing potency and safety[62]. In this report we offer a metabolic approach utilizing a glucose transporter already found in nature to improve upon the current standard of CARs. The data further provide fundamental insights into the crosstalk between nutrient signaling and metabolic and transcriptional programming in human T cells.

## Methods

### Human CAR T cell production
Healthy donor buffy coats were purchased from the Stanford Blood Center under an IRB-exempt protocol. Primary human T cells were isolated using the RosetteSep Human T cell Enrichment kit (Stem Cell Technologies) according to the manufacturer's protocol. Isolated T cells were cryopreserved in CryoStor CS10 cryopreservation medium (Stem Cell Technologies).

Non-tissue culture treated 12-well plates were coated overnight at 4 °C with 1 ml Retronectin (Takara) at 25 µg/ml in PBS. Plates were washed with PBS and blocked with 2% BSA for 15 min. Thawed or fresh retroviral supernatant was added at approximately 1 ml per well and centrifuged for 2 h at 32 °C at 2400 × $g$ before the addition of cells. Primary human T cells were thawed and activated with Human T-Expander CD3/CD28 Dynabeads (Gibco) at a 3:1 bead:cell ratio in complete medium (RPMI 1640 supplemented with 10% fetal bovine serum, 10 mM N-2-hydroxyethylpiperazine-N9-2-ethanesulfonic acid, 2 mM GLUTaMAX, 100 U/mL penicillin (Gibco), and 100 U/mL of IL-2 (Peprotech)). T cells were transduced with retroviral vector on day 2 post-activation. Beads were taken off at day 4 post-activation.

### Mice
Immunodeficient NSG mice (NOD.Cg-Prkdc$^{scid}$Il2rg$^{tm1Wjl}$/SzJl) were bred in house. Mice used for in vivo experiments were between 6 and 10 weeks old, and the ratio of male to female mice was matched in experimental and control groups. All animal studies were carried out according to Stanford University Animal Care and Use Committee–approved protocols in a barrier facility. Facilities contained standard day/light cycles with ambient temperature and humidity. Maximum tumor burden was determined by bioluminescent flux values at or greater than 1 × 10$^9$ photons per second. End point euthanasia performed by exposure to CO$_2$.

Nalm6 Challenge: Female mice were inoculated with 1 × 10$^6$ Nalm6-GL leukemia via intravenous injections. All CAR T cells were injected intravenously at day 14 post-activation. Bioluminescence imaging was performed using a Spectrum IVIS instrument. Values were analyzed using Living Image software. Mice were humanely euthanized when an IACUC-approved end-point when mice demonstrated signs of morbidity and/or hind-limb paralysis (leukemia). Mice were randomized to ensure equal pre-treatment tumor burden before CAR T cell treatment.

Circulating CAR T cells were identified using anti-human CD45 (BD Biosciences) and CountBright beads (Thermo Fisher).

### Seahorse assays for ECAR and OCR
Metabolic analyses were carried out using Seahorse Bioscience Analyzer XFe96. Briefly, 0.2 × 10$^6$ cells were resuspended in extracellular flux (XF) assay media supplemented with 25 mM glucose, 2 mM glutamine, and 1 mM sodium pyruvate and plated on a Cell-Tak (Corning)–coated microplate allowing the adhesion of CAR T cells. Mitochondrial activity and glycolytic parameters were measured by the oxygen consumption rate (OCR) (pmol/min) and extracellular acidification rate (ECAR) (mpH/min), respectively, with use of real-time injections of oligomycin (1.5 mM), carbonyl cyanide ptrifluoromethoxyphenylhydrazone (FCCP; 0.5 mM), and rotenone and antimycin (both at 0.5 mM). In-Seahorse activation was performed at day 16 post-activation, using 5 mg/mL of idiotype crosslinked with 10 mg/mL of mouse anti-F(ab')2 (Jackson ImmunoResearch). Respiratory parameters were calculated according to manufacturer's instructions (Seahorse Bioscience). All chemicals were purchased from Agilent unless stated otherwise.

### Bulk RNAseq
For bulk RNA isolation, healthy donor T cells were prepared as described above. On days 14–16 control cells or GLUT1 overexpressing CAR-T cells were collected and total mRNA was isolated using Qiagen RNEasy Plus mini isolation kit. Bulk RNAseq was performed by Novogene using the NovaSeq platform. The raw RNA sequencing data was mapped to human reference genome hg38 using the STAR aligner[63], and genes annotated in Gencode v36[64] was quantified using featurecounts in the subread package[65]. The differential gene expression analysis was conducted in the DESeq2 package[66]. Gene set enrichment analysis was performed with Gene Set Enrichment Analysis[67]. Data visualization is generated using python. Significantly different genes were identified by DESeq2 using Wald test. DAVID gene annotation enrichment analysis was performed using KEGG pathways and GO terms (biological process, cellular component, and molecular function). Functional annotation clustering was performed and terms with $p < 0.05$ (Benjamini corrected) are shown. Redundant terms were manually removed for visualization.

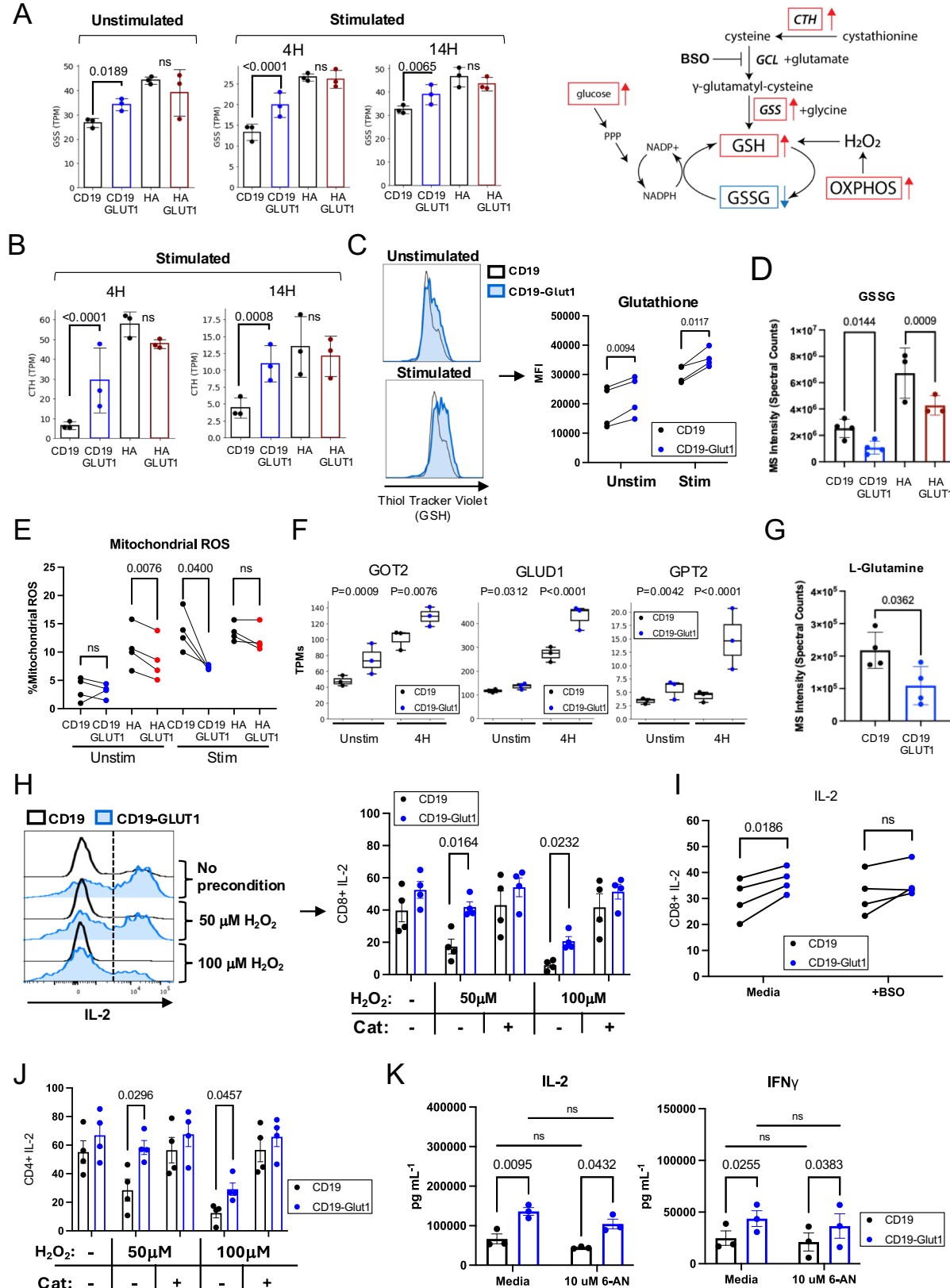

## Mass cytometry

$1 \times 10^6$ cells were washed with two times with PBS and then resuspended in 1 ml of 250 nM cisplatin and PBS (Fluidigm) for assessing cell viability. Cells were incubated for 3 min at RT and washed with cell staining medium (CSM, 1X PBS with 0.05% BSA, 0.02% sodium azide). Cells were fixed with 1.6% paraformaldehyde diluted in PBS for 10 min at RT, then washed with PBS. Samples were subsequently frozen using CryoStor-10 (Thermo Fisher Scientific) for further use. Upon thawing and washing in CSM, barcoding was performed and samples were pooled[68]. A master mix of titrated surface antibodies

**Fig. 5 | GLUT1 overexpression enhances metabolic pathways that favor resistance to REDOX suppression. A** (LEFT) Transcripts Per Million (TPMs) of GSS (Glutathione Synthetase) transcripts for CD19 and HA ± GLUT1OE CAR-T cells at baseline and stimulated for 4 h and 14 h with anti-idiotype. Data from *n* = 3 donors. *P* values calculated using DESeq2. (RIGHT) Schematic of pathways involved in GSH REDOX with relevant metabolomic derivatives. Red arrows indicate elements found to be enriched with GLUT1OE. Blue arrows indicate elements found to be decreased with GLUT1OE. **B** TPMs of CTH (cystathionine gamma-lyase) transcripts for CD19 and HA ± GLUT1OE CAR-T cells stimulated for (LEFT) 4 h and (RIGHT) 14 h with anti-idiotype. Data from *n* = 3 donors. *P* values calculated using DESeq2. **C** (LEFT) Representative histograms showing CAR⁺ intracellular GSH content using ThiolTracker for CD19 ± GLUT1OE CAR-T cells ± 4 h 1 μg / mL anti-idiotype stimulation. (RIGHT) Quantitative data of CAR + GSH MFI. Cotransduced cells were magnetically enriched for greater than 95% double positive prior to experiment. Data from *n* = 4 donors. *P* values determined by paired *t*-tests. **D** Untargeted LC-MS data depicting cysteineglutathione disulfide (GSSG, or oxidized GSH) abundance in electronically sorted CD19 and HA ± GLUT1OE CAR-T cells at day 14. Pooled data of *n* = 4 donors. *P* values determined by unpaired two-tailed *t*-tests. Error bars represent SD. **E** Pooled mitochondrial ROS data collected via intracellular staining. Day 15 CAR-T cells were subject to 2 h anti-idiotype stimulation. Data were reflective of *n* = 4 donors. *P* values determined by paired *t*-tests. **F** A TPMs of glutaminolysis-related genes GOT2, GLUD1, and GPT2 transcripts for CD19 ± GLUT1OE CAR-T cells unstimulated or stimulated for 4 h with anti-idiotype. Data from *n* = 3 donors. *P* values calculated using DESeq2. **G** Untargeted LC-MS data depicting L-glutamine abundance in electronically sorted CD19 ± GLUT1OE CAR-T cells at day 14. Pooled data of *n* = 4 donors. *P* values determined by unpaired two-tailed *t*-tests. Error bars represent SD. **H** Intracellular IL-2 staining for CD8⁺ CD19 ± GLUT1OE CAR-T cells ± pre-exposure to oxidative stress (hydrogen peroxide). (LEFT) Representative flow cytometry. (RIGHT) Pooled data. Cells were challenged with Nalm6-GL at a 1:1 ratio on day 14. Data from *n* = 4 donors. *P* values determined by paired *t*-tests. Error bars represent SEM. **I** Intracellular IL-2 staining for CD8⁺ CD19 ± GLUT1OE CAR-T cells stimulated ± BSO (Buthionine Sulfoximine). Cells were stimulated via 1 ug/L plate-bound anti-idiotype on day 14. Data from *n* = 4 donors. *P* values determined by paired *t*-tests. Error bars represent SEM. **J** Intracellular IL-2 staining for CD4⁺ CD19 ± GLUT1OE CAR-T cells ± pre-exposure to oxidative stress (hydrogen peroxide). Cells were challenged with Nalm6-GL at a 1:1 ratio on day 14. Data from *n* = 4 donors. *P* values determined by paired *t*-tests. Error bars represent SEM. **K** Cytokine secretion of CD19 ± GLUT1OE CAR-T cells challenged 1:1 with Nalm6 leukemia ± 6-Aminonicotinamide (6-AN) on day 14 as measured by ELISA. Data reflective of *n* = 3 donors. *P* values determined by paired *t*-tests. Error bars represent SEM.

was prepared and filtered (0.1 mm) then added to pooled and barcoded sample for 30 min at RT. Following surface staining, samples were washed twice in CSM and permeabilized with ice cold methanol for 10 min on ice, then washed again twice in CSM. Samples were stained with titered intracellular antibodies for 45 min on ice followed by 2 subsequent washes with CSM. Finally, samples were resuspended in DNA intercalator (Fluidigm, 1:5000 191Ir/193Ir and 1% PFA in 1X PBS) and incubated overnight at 4 °C for next day acquisition (Helios). On day of acquisition, samples were washed once in CSM and twice in filtered ddH₂O. Cells were then resuspended at $1 \times 10^6$ cells/ml in ddH₂O with 1x EQ four-element beads (Fluidigm Corporation, no. 201078). Cells were acquired on a Fluidigm Helios mass cytometer.

Data was analyzed using OMIQ software.

## Untargeted metabolomics by LC-MS

Metabolites were extracted from CAR-T cells and analyzed using a broad spectrum LC–MS platform as previously described (Contrepois et al., MCP 2015, Contrepois et al., Cell 2020).

A solvent mixture of 80:20 methanol/water (500 μl) containing seven internal standards was used to resuspend the pellets that were composed of $2$–$4 \times 10^6$ cells. Cell suspensions were vortexed for 30 s, sonicated in a water bath (30 s sonication, 30 s on ice, repeated 3 times), vortexed for 30 s and incubated for 2 h at −20 °C to allow protein precipitation. The supernatant was collected after centrifugation at 10,000 r.p.m. for 10 min at 4 °C and evaporated to dryness under nitrogen. The dry extracts were then reconstituted with 100 μl of 50:50 methanol/water before analysis.

Polar metabolites were analyzed using HILIC separation in both positive and negative ionization modes. HILIC experiments were performed using a ZIC-HILIC column (2.1 × 100 mm, 3.5 μm, 200 Å; Merck Millipore) and mobile phase solvents consisting of 10 mM ammonium acetate in 50:50 acetonitrile:water (A) and 10 mM ammonium acetate in 95:5 acetonitrile:water (B). Data were acquired on a Q Exactive HF Hybrid Quadrupole-Orbitrap mass spectrometer equipped with a HESI-II probe and operated in full MS scan mode. MS/MS data were acquired on a pool sample. Data quality was ensured by (1) sample randomization for metabolite extraction and data acquisition, (2) injection of 12 pool samples to equilibrate the LC–MS system before running the sequence, (3) injection of pool samples every 10 injections to control for signal deviation with time, and (4) checking mass accuracy, retention time, and peak shape of internal standards in every samples.

Data were processed using Progenesis QI software (v2.3) (Nonlinear Dynamics, Durham, NC). Metabolic features from blanks and those that did not show sufficient linearity upon dilution in QC samples ($r < 0.6$) were discarded. Only metabolic features present in >2/3 of the samples were kept for further analysis. Metabolite abundances were normalized using total protein content as measured by BCA Protein Assay Kit on the protein pellet (Pierce). Missing values were imputed by drawing from a random distribution of low values in the corresponding sample. Data from each ionization mode were merged and metabolites were annotated using authentic standards and publicly available MS2 databases. We used the Metabolomics Standards Initiative (MSI) level of confidence to grade metabolite annotation confidence and reported metabolites with levels 1 and 2.

## Flow cytometry

All flow analysis was performed at day 14/15 post-T cell activation, unless differently indicated in the text. The anti-CD19 CAR idiotype antibody was provided by B. Jena and L.Cooper. The 1A7 anti-14G2a idiotype antibody was obtained from NCI Frederick and University of Texas M.D. Anderson Cancer Center. The anti-idiotype antibodies and Fc-fusion protein were conjugated in house with Dylight650 antibody labeling kits (Thermo Fisher). Surface GLUT1 expression was monitored as a function of binding to its ligand, the envelope glycoprotein of the human T lymphotrophic virus (HTLV). A recombinant HTLV envelope receptor binding domain (HRBD) fused to the EGFP coding sequence was used as previously described (Manel et al., 2003; Montel-Hagen et al., 2008b).

Analysis of GLUT1OE CAR T cells performed by gating on CAR and NGFR double positive populations (percentage of double positive >80% for each experiment unless otherwise noted).

T cell phenotype was assessed using the following antibodies at 1:50 dilution unless otherwise noted:

From METAFORA: GLUT1-specific H2RBD-GFP ligand (anti-GLUT1 ligand) (1:100).

From BioLegend: CD4-APC-Cy7 (clone OKT4), CD8-PerCp-Cy5.5 (clone SK1), TIM-3-BV510 (clone F38-2E2), CD39-FITC, PE or APC-Cy7 (clone A1), IL-2-PE/Cy7 (clone MQ1-17H12), CD62L-BV605 (clone DREG-

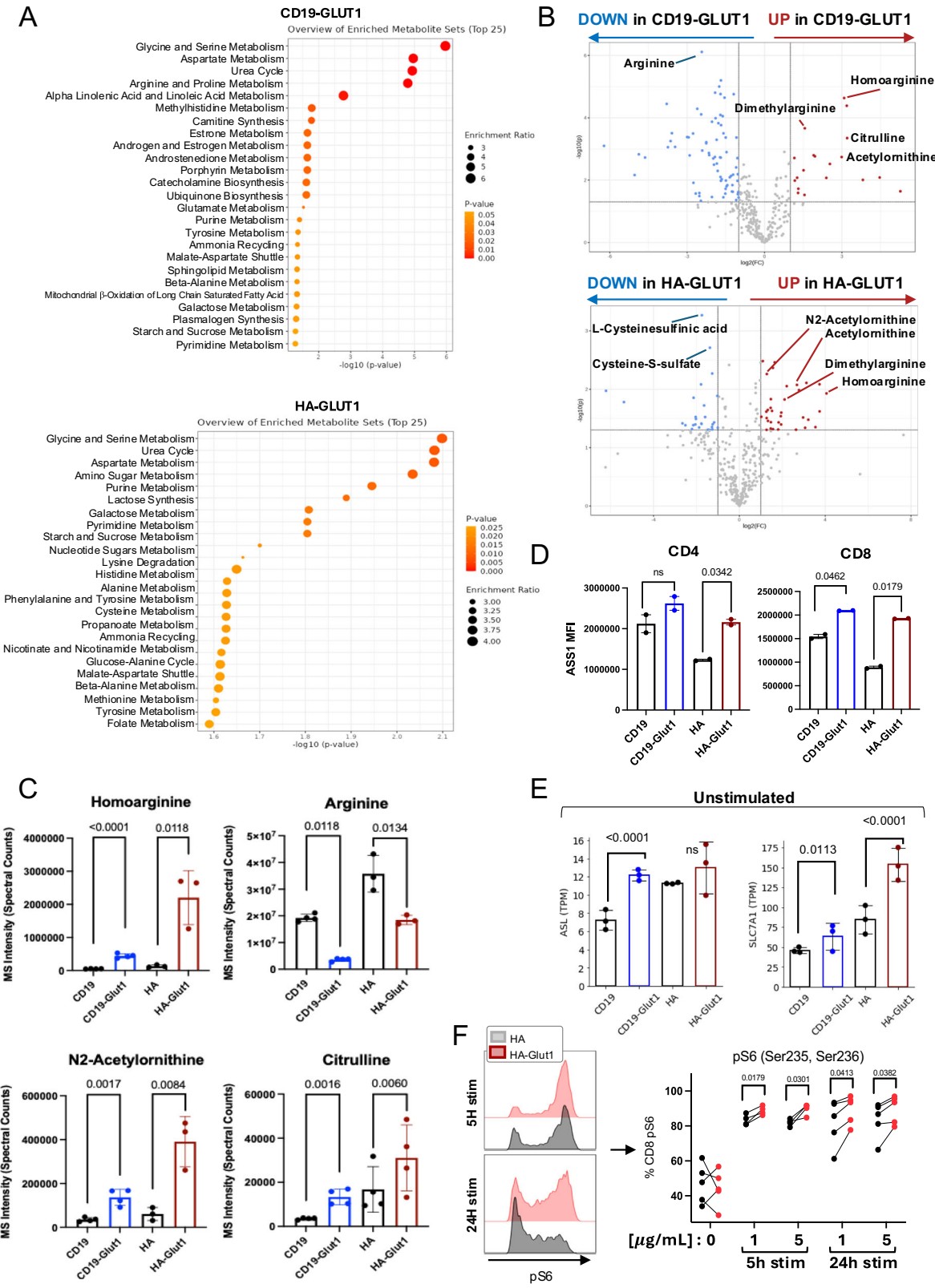

56), CD45RO-PE/Cy7 (clone UCHL1), CCR6-BV421 (clone G034E3), CCR4-PE (clone L291H4).

From eBioscience: PD-1-PE-Cy7 (clone eBio J105), LAG-3-PE (clone 3DS223H), CD45RO-PE-Cy7 (clone UCHL1), CD45-PerCp-Cy5.5 (clone HI30).

From BD: LAG-3-BV421 (clone T47-530), CD62L-BV605 (clone DREG-56), CD4-BUV395 (clone SK3), CD8-BUV805 (clone SK1), BrdU-PerCP-Cy5.5 (clone 3D4), CD271-BUV737 (clone C40-1457), Fixable Viability Stain 510 (0.4:100).

From Thermo Fisher Scientific: Phospho-S6-PE/Cy7 (clone cupk43k), Thioltracker Violet (glutathione detection reagent).

Data was collected with an LSR Fortessa X-20 (BD Bioscience) or Cytek Aurora (Cytek Biosciences) and analyzed using FlowJo software.

**Fig. 6 | GLUT1 overexpression alters arginine metabolism. A** LC-MS data depicting top metabolomic pathways enriched in electronically sorted (TOP) CD19-GLUT1 and (BOTTOM) HA-GLUT1 CAR-T cells on day 14. Significantly differential metabolites were analyzed using MetaboAnalyst. **B** Volcano plots of metabolite abundance related to the Urea Cycle in electronically sorted (TOP) CD19-GLUT1 and (BOTTOM) HA-GLUT1 CAR-T cells on day 14. Red circles indicate metabolites that significantly increased. Blue circles indicate metabolites that significantly decreased. Data from $n = 4$ donors. **C** Quantitative data showing metabolites involved in Urea Cycle for electronically sorted CD19 and HA ± GLUT1OE on day 14. Data from $n = 3$ or 4 donors. *P* values determined by unpaired two-tailed *t*-tests. Error bars represent SD. **D** Flow cytometric analysis of intracellular expression of Argininosuccinate synthase 1 (ASS1) on (LEFT) CD4 and (RIGHT) CD8 CD19 and HA ± GLUT1OE CAR-T cells on day 16. Quantitative data from $n = 2$ donors. *P* values determined by paired two-tailed *t*-tests. Error bars represent SD. **E** TPMs of (LEFT) ASL (argininosuccinate lyase) and (RIGHT) SLC7A1 transcripts for CD19 and HA ± GLUT1OE CAR-T cells at baseline. Data from $n = 3$ donors. *P* values determined by paired two-tailed *t*-tests. Error bars represent SD. **F** (LEFT) Representative histograms of intracellular phosphorylated ribosomal subunit 6 (pS6) 5 and 24 h after anti-idiotype stimulation for CD8 + HA ± GLUT1OE CAR-T cells on day 16. (RIGHT) Quantitative data of CD8⁺ pS6⁺ HA ± GLUT1OE CAR-T cells ± stimulation. Data from $n = 4$ or 5 donors. *P* values determined by paired two-tailed *t*-tests.

## [U13C] glucose tracing

CAR-T cells were resuspended in glucose free media and serum (5%) for 1 h at 37 °C. $2 \times 10^6$ cells were plated with 11 mM glucose or [U13C] glucose ± idiotype stimulation for 4 h at 37 °C. Harvested cells were spun down, resuspended in ammonium carbonate, spun down, and resuspended in lysis buffer. Lysates were spun down at $18,000 \times g$ for 10 min and placed at −80 °C.

## [U13C] LC-MS/MS Data Acquisition

Metabolite extracts arising from the isotopic labeling experiment outlined above were further processed for LC-MS/MS and analyzed by General Metabolics (Cambridge, MA). 6 μL of each sample was injected and separated by UHPLC using a Nexera UHPLC system (DGU-405 degasser unit, LC40DX3 solvent delivery system, SIL-40CX3 auto sampler, CBM-40 system controller, CTO-40C column oven; Shimadzu). Separation was achieved by hydrophilic interaction liquid chromatography (HILIC) using an Atlantis Premier BEH Z-HILIC column (1.7 μm, 2.1 × 50 mm; 186009978, Waters). Separation was achieved using a 5 min multi-phase linear gradient of the following buffers: Buffer A) 9:1 acetonitrile:water (v:v) 10 mM ammonium acetate pH 9.2, B: 1:9 acetonitrile:water (v:v) 10 mM ammonium acetate pH 9.2. Samples were ionized using an Optimus Turbo V + Dual TIS ion source (S ciex) and were analyzed using an X500R mass spectrometer (Sciex). Samples were analyzed in negative ionization mode and were acquired using data-dependent acquisition. MS1 data was acquired with a 0.2 s accumulation followed by top 5 MS2 data acquisition with an accumulation time of 0.04 seconds per target. Dynamic background subtraction was used and former candidate ions were excluded for 10 s. A collision energy of −15 V was used with a CE spread of 10.

## [U13C] metabolic tracing analysis

LC-MS/MS data processing and ion annotation was performed according to accepted protocols for mass spectrometry data processing and metabolite annotation[69], as well as the comparison of isotopic labeling patterns. Briefly, annotation was based on matching of chromatographic retention times and MS1 values from detected features to those of measured purified standards and/or matching of the resulting MS2 spectra from fragmentation to spectral libraries of authentic fragmented metabolite standards. This matching was performed by comparison with signals from control samples that were not exposed to metabolic labeling. The expected m/z values for isotopologues of identified features were generated and those features were annotated based on the observed retention time for the unlabeled M + 0 mass to the expected MS1 values for those isotopologues across the remaining labeled samples in the dataset. Mass distribution vectors (MDVs)[70] were calculated per sample measured and based on the ratio of the feature height detected for the indicated isotopologue compared to the sum of the feature heights for all detected isotopologues for that annotated metabolite. These analyses and calculations were executed at General Metabolics, a commercial metabolomics service vendor.

## 2-NBDG staining

$0.1 \times 10^6$ CAR-T cells were plated in 96-well plates with and without plate bound coated idiotype and stimulated for 24H. For the last 30 min, cells were incubated in no glucose RPMI 1640 and then for another 30 min 50 μM of 2-NBDG was added (abcam 235976) was performed accordingly to the manufacturer protocol.

## In vitro glucose uptake

Glucose uptake for $0.3 \times 10^6$ CAR T cells was measured by 2-Deoxy-D-[1,2-3H (N)]-glucose uptake as described[71]. Briefly, cells were washed 3 times with the KRH buffer and subjected to serum starvation in KRH buffer supplemented with 0.2% BSA for 2–3 h. For measurement of glucose uptake, CAR T cells in both stimulated and unstimulated stages were incubated with 2-Deoxy-D-[1,2-3H (N)]-glucose (1.71μCimL-1) along with 500 uM of 2-deoxy-D-glucose for 10 min. Cells were immediately washed 3 times with ice-cold PBS followed by cell lysis in 0.1% SDS buffer (w/v in water). A small amount of lysate (90 μl) was used to count the radioactivity in the liquid scintillation counter. Deoxy-D-[1,2-3H (N)]-glucose was purchased from Perkin Elmer which is now Revvity (catalog number, NET328A001MC).

## Cytokine production assays

T cells and tumor cells (E:T as specified in figure legends) were cocultured in 250 μl media without IL-2 in round bottom 96-well plates for 24 h. Culture supernatants were collected and analyzed by enzyme-linked immunosorbent assay (ELISA). IL-2 and IFNγ were detected with the ELISA MAX kit (Biolegend), and TNFα was detected with the Quantikine kit (R&D Systems). Bead-based multiplex cytokine detection assays were performed at the Human Immune Monitoring Center (Stanford University) using the Luminex platform. Mock T cells, incubated with respective tumor, were included as a negative control.

## In vitro staining for intracellular glutathione content

$0.1 \times 10^6$ CAR-T cells were resuspended in RPMI containing serum and 1.1 mM glucose 14 days post activation. Cells were stimulated using 1 mg/mL plate bound idiotype for 4 h, washed, then stained with 5 mM ThiolTracker Violet for 20 min in DPBS containing calcium/magnesium/glucose/pyruvate (Thermo Scientific). Both stimulated and unstimulated cells were fixed overnight at 4 °C and stained for flow cytometry analysis the following day.

## In vitro intracellular staining for IL2 in the presence of BSO

$0.2 \times 10^6$ CAR-T cells were plated in 96-well plates with and without plate bound coated idiotype. Cells were resuspended in IL-2-free AIMV media with 5% serum or AIMV containing 4 mM BSO (Sigma Aldrich).

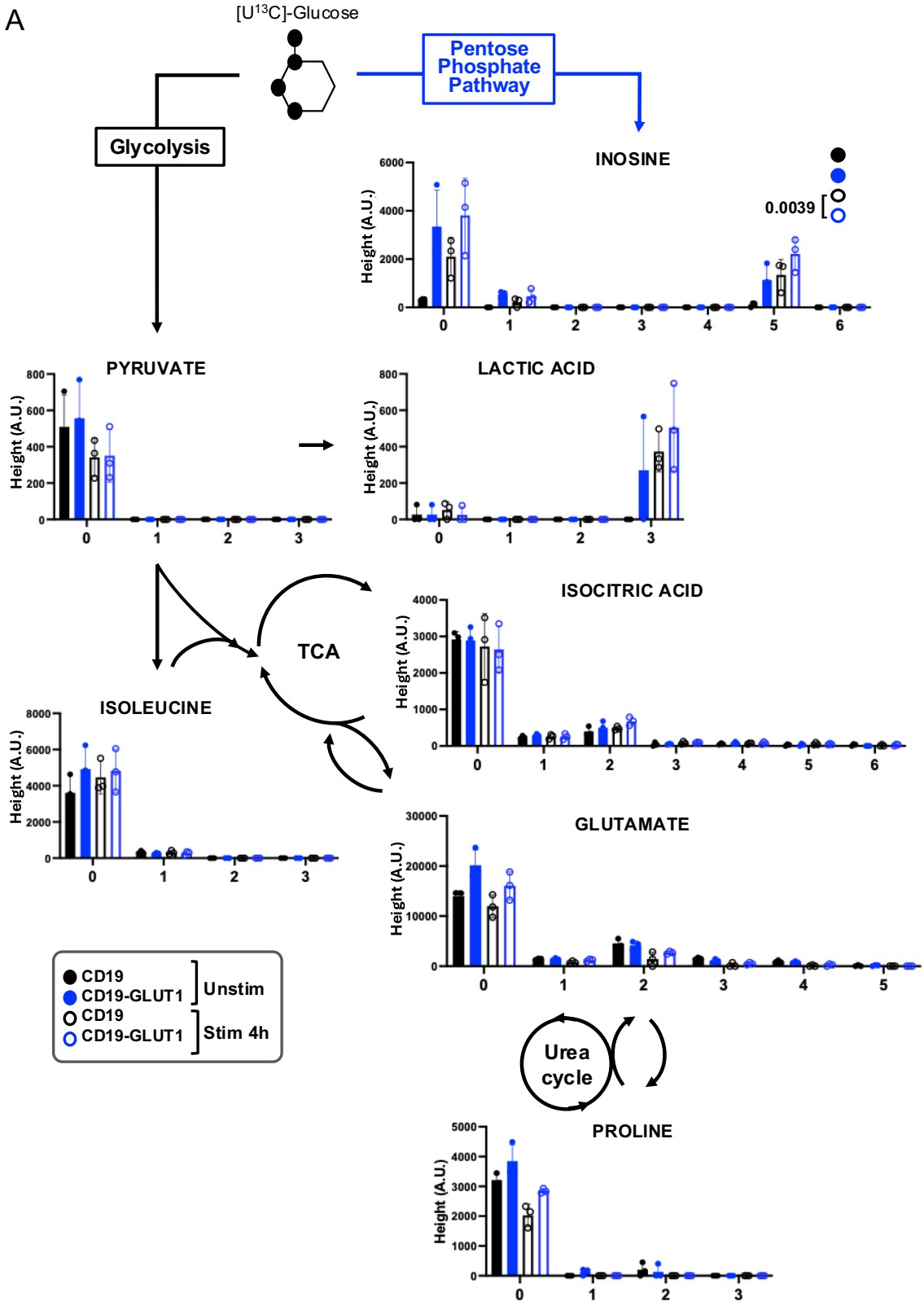

**Fig. 7 | 13C-Glucose tracing in CD19 CAR-T cells with GLUT1OE. A** [U13C] glucose tracing. CAR-T cells ± GLUT1OE were administered 11 mM labeled glucose ± 4 h idiotype stimulation. X axis reflects isotopologue. Y axis represents Height in aleatory units. Statistics generated by paired, two tailed t-test (90% confidence) based on sum of total isotopologues ≥ 1 in each condition. Data from n = 3 donors. Error bars represent SD.

Cells were treated with 1x Protein Transport Inhibitor Cocktail (eBiosciences) at start of culture and stimulated for 6 h. Fixation and permeabilization for analysis of IL-2 were done following the manufacturer's protocol (eBiosciences).

**In vitro intracellular ROS suppression assay**

$1 \times 10^6$ CAR+ cells were exposed to 50 or 100 mM hydrogen peroxide (Sigma Aldrich) for 1 h at 37 °C in RPMI without IL-2. As a control, some cells were also given 100 U/mL catalase just prior to

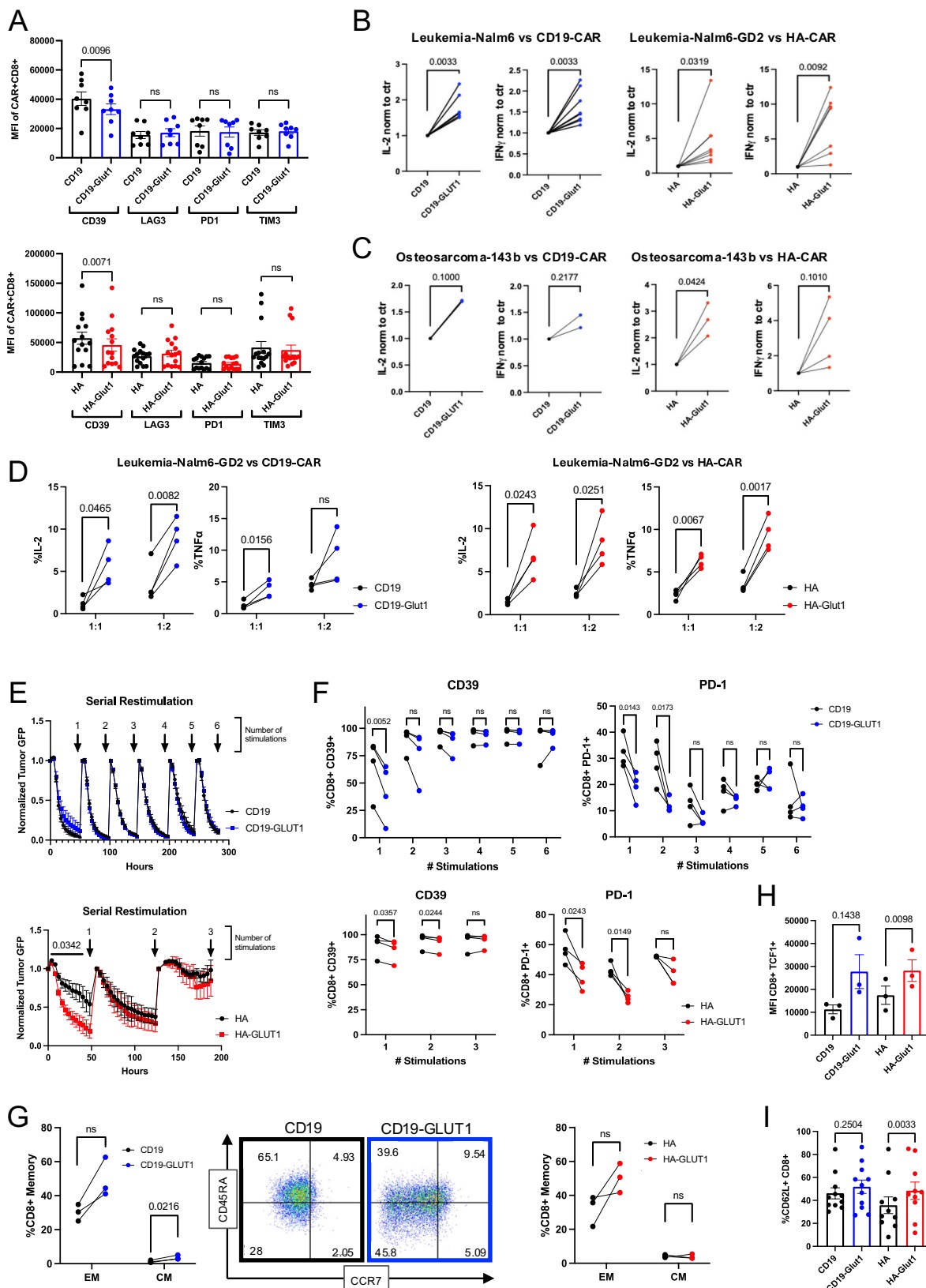

exposure to hydrogen peroxide. Cells were subsequently spun down to remove hydrogen peroxide and/or catalase from the media. $1 \times 10^5$ CAR+ cells were replated for challenge with $1 \times 10^5$ Nalm6-GD2 for 6 h in a 96 well plate. Cells were treated

with 1x Protein Transport Inhibitor Cocktail (eBiosciences) 2 h after start of culture. Fixation and permeabilization for analysis of IL-2 were done following the manufacturer's protocol (eBiosciences).

**Fig. 8 | GLUT1 overexpression enhances potency and delays onset of exhaustion in response to tumor rechallenge. A** MFI of exhaustion markers CD39, LAG3, PD1 and TIM3 expressed by (TOP) CD19 or (BOTTOM) HA ± CD8⁺ GLUT1OE CAR-T cells on day 14. Pooled data of $n = 14$ donors. *P* values determined by paired two-tailed *t*-tests. Error bars represent SEM. **B** ELISA analysis of IL-2 and IFNγ secretion by CD19 and HA ± GLUT1OE CAR-T cells after 24 hour stimulation with (LEFT) Nalm6 or (RIGHT) Nalm6-GD2 leukemia tumor lines on day 14. Data from $n = 6$ donors. *P* values determined by paired two-tailed *t*-tests. **C** ELISA analysis of IL-2 and IFNγ secreted by CAR-T cells ± GLUT1OE CAR-T cells after 24 hour stimulation with 143b against (LEFT) CD19 and (RIGHT) HA on day 14. Data from $n = 2$–4 donors. *P* values determined by paired two-tailed *t*-tests. **D** Intracellular staining of IL-2 and TNFα by (LEFT) CD19 and (RIGHT) HA ± GLUT1OE CAR-T cells after 24 h of stimulation with Nalm6-GD2 leukemia at 1:1 and 1:2 E:T on day 14. Data from $n = 4$ donors. *P* values determined by paired two-tailed *t*-tests. **E** Serial rechallenge and tumor-GFP killing kinetics data using Incucyte. Pooled data of 4 donors (TOP) CD19 and

(BOTTOM) HA ± GLUT1OE CAR-T cells sequentially challenged at 1:2 ratio with Nalm6 leukemia ± GD2. Incucyte *p* values generated using two way ANOVA. Error bars represent SEM. **F** Flow cytometry measurements of CD39 and PD-1 of CD8⁺ (TOP) CD19 and (BOTTOM) HA ± GLUT1OE CAR-T cells after each stimulation denoted by arrows in (**E**). *P* values determined by paired two-tailed *t*-tests. **G** Memory formation data of serially rechallenged CD8⁺ CAR-T cells after (LEFT) 4 stimulations for CD19 ± GLUT1OE with representative flow cytometry of CD62L⁺ cells and (RIGHT) 3 stimulations for HA ± GLUT1OE. EM- effector memory, CM- central memory. *P* values determined by paired two-tailed *t*-tests. **H** Pooled intracellular expression of TCF1 in CD19 and HA ± GLUT1OE CAR-T cells on day 16. Data from $n = 3$–4 donors. *P* values determined by paired two-tailed *t*-tests. Error bars represent SEM. **I** Pooled surface expression of CD62L in CD19 or HA ± GLUT1OE CAR-T cells on day 16. *P* values determined by paired two-tailed *t*-tests. Error bars represent SEM.

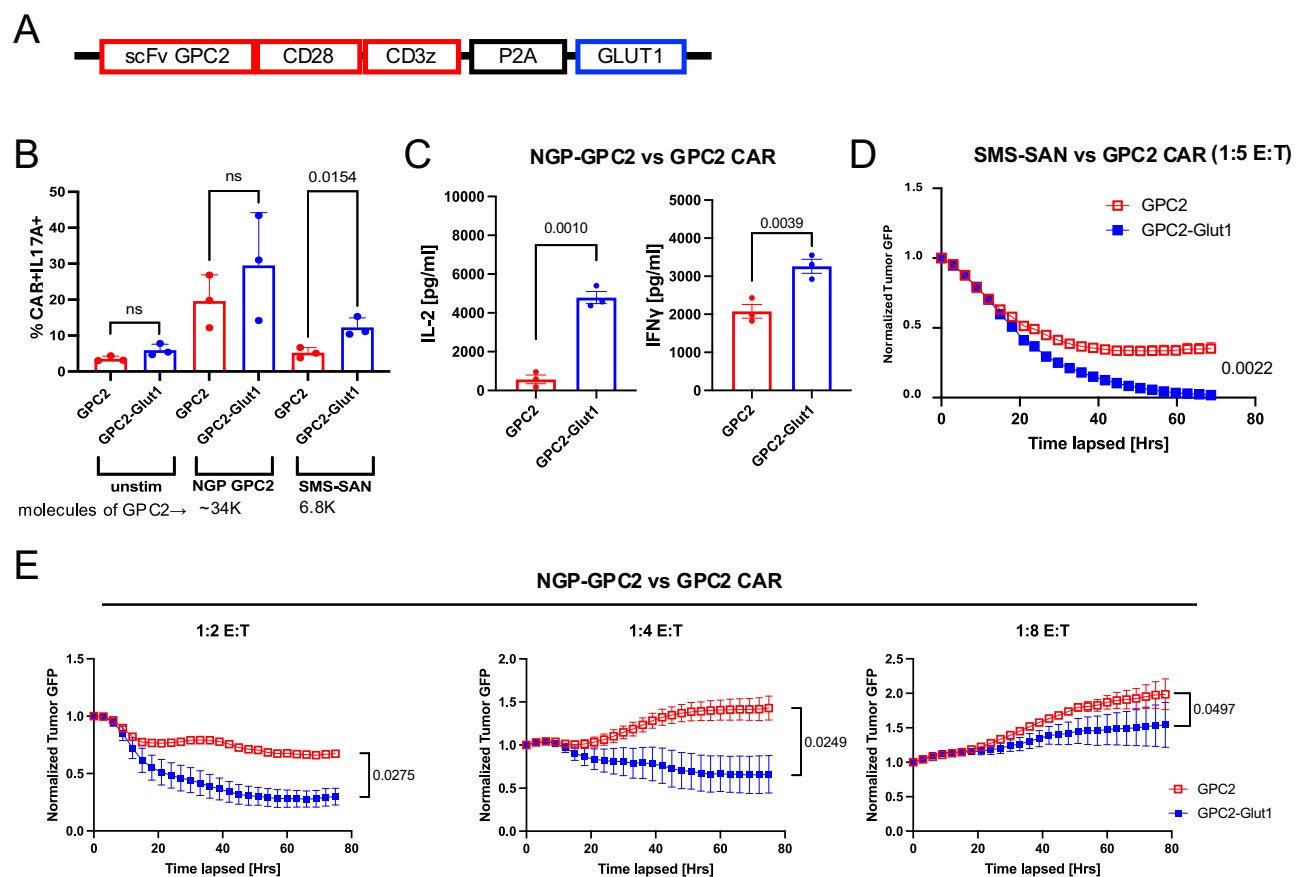

**Fig. 9 | GLUT1 overexpression increases potency of GPC2-CAR T cells against neuroblastoma in vitro. A** Design of GPC2 CAR and GLUT1 expressing vector (**B**) Intracellular cytokine staining after 24 h of GPC2 ± GLUT1OE CAR T cells stimulated with three different tumor lines on day 14. Error bars represent mean ± SD of triplicate wells from one donor. *P* values determined by unpaired two-tailed *t*-tests. **C** Day 14 post activation CAR T cells stimulated with NGP-GPC2 at 1:1 E:T ratio. IL-2 and IFNγ secretion was assessed 24 h post-stimulation via ELISA. Data from $n = 3$

donors. *P* values determined by paired two-tailed *t*-tests. Error bars represent SEM. **D** Day 14 post-activation CAR T cells stimulated with SMS-SAN-GL tumor line at 1:5 E:T ratio and tumor killing was assessed using Incucyte. Error bars represent mean ± SD of triplicate wells from one representative donor ($n = 2$ donors). *P* values determined by two-way ANOVA. **E** Tumor killing kinetics of GPC2 ± GLUT1OE CAR T cells challenged with NGP-GPC2 across E:Ts captured by Incucyte. *P* values determined by two-way ANOVA.

## Phospho-flow for pS6 after idiotype stimulation

Stimulation was achieved by coating non tissue cultured 24 well plates with either 1 mg/mL or 5 μg/mL IA7 idiotype in PBS overnight. $1 \times 10^6$ HA or HA-GLUT1 CAR-T cells were stimulated 15 days post-activation after plate blocking with 2% BSA in PBS. Cells were cultured in RPMI without IL-2 for either 6 or 24 h, harvested, and stained for surface markers. Cells were fixed and permeabilized as per manufacture's

protocol (Invitrogen) with the addition ice cold methanol introduced during fixation.

## Co-culture assays with 6-AN

Control or overexpressing GLUT1 CAR-T cells at day 14 post-activation were cultured with Nalm6 tumor at the 1:1 ratio in the presence of 10 μM 6-AN resuspended in DMSO. Cell culture

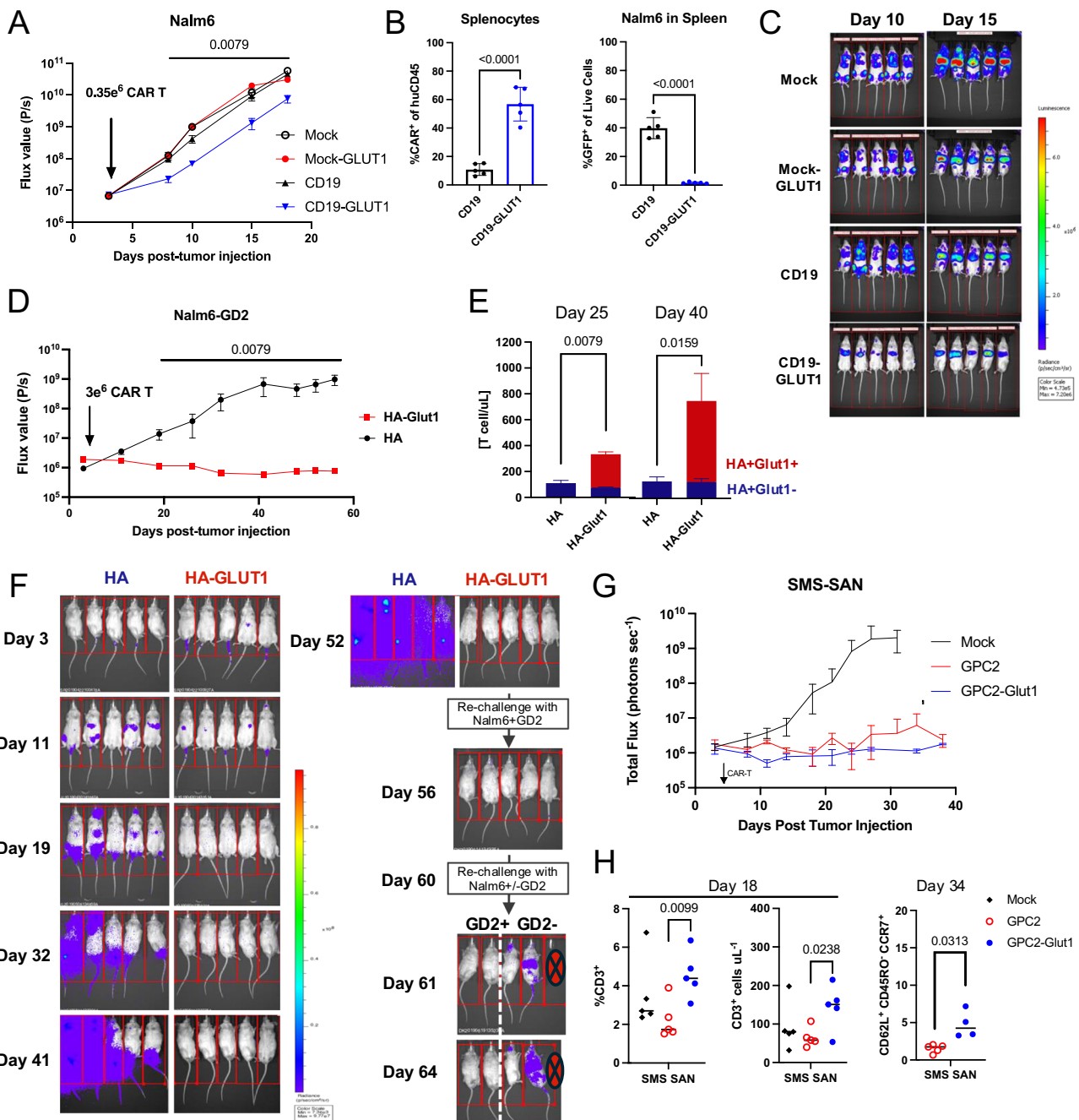

**Fig. 10 | GLUT1 overexpression enhances CAR-T cell tumor clearance in vivo.**
**A** Tumor progression was monitored using bioluminescent imaging. Data are mean ± SD of $n = 5$ mice per group. Data representative of $n = 2$ experiments. *P* values determined by Mann-Whitney test. **B** Flow cytometry analysis of total splenocytes for (LEFT) %CAR⁺ cells of human CD45 and (RIGHT) Nalm6 GFP. Statistics generated by unpaired two-tailed *t*-tests reflective of $n = 5$ mice per group. **C** Representative BLI of tumor progression in vivo. **D** Tumor progression was monitored using bioluminescent imaging. Data are mean ± SD of $n = 5$ mice per group. Data representative of $n = 3$ experiments. *P* values determined by Mann-Whitney test. **E** T cells detected in peripheral blood at (TOP) 25 and (BOTTOM) 40 days post-tumor injection. Data are mean ± SEM of $n = 5$ mice per group. **F** BLI of tumor progression in vivo. At days 52 and 60 post-tumor injection, HA-GLUT1 CAR-T cells were re-challenged. On day 60, HA-GLUT1 cohort were challenged with either Nalm6-GD2 or Nalm6 tumor cells. **G** Tumor progression was monitored using bioluminescent imaging. Data are mean ± SD of $n = 5$ mice per group. Data representative of $n = 3$ experiments. *P* values determined by Mann-Whitney test. **H** Flow cytometry analysis of peripheral blood for (LEFT) %CD3⁺ of live cells on day 18 and (RIGHT) $T_{SCM}$ populations on day 34 post tumor injection. Statistics generated by unpaired two-tailed t-tests reflective of $n = 5$ mice per group. Data representative of $n = 3$ experiments. Error bars represent SD.

supernatants were collected at 24 h, and interleukin-2 (IL-2) and interferon-g concentrations were determined by enzyme-linked immunosorbent assay (BioLegend). Triplicate wells were plated for each condition.

**Staining for mitochondrial ROS**

$0.3 \times 10^6$ CAR T cells were subject to ± stimulation (2 h) in RPMI without IL-2. Cells were stained as per manufacturer's protocol (Cayman Chemical 701600) for 20 min in mitochondrial detection reagent at a

concentration of 0.5 uM. After subsequent washes cells were analyzed using BD Fortessa on the PE channel.

### Serial restimulation of CAR-T cells
Tumor killing kinetics of $0.05 \times 10^6$ CAR-T cells challenged against $0.1 \times 10^6$ GFP$^+$ tumor was monitored using Incucyte instrument. Upon observation of tumor clearance cells were counted and replated again at a 1:2 E:T and stained for CD39 and PD-1.

### Statistical analysis and graphical design
Unless otherwise noted, statistical analyses for significant differences between groups were conducted using unpaired two-tailed $t$-tests without correction for multiple comparisons and without assuming consistent s.d. using GraphPad Prism 10. Graphical abstract and schematics were designed in Adobe Illustrator.

### Reporting summary
Further information on research design is available in the Nature Portfolio Reporting Summary linked to this article.

## Data availability
The RNA-seq data generated in this study have been deposited in the GEO database under accession code GSE275152 and can be found at: Source data are provided with this paper.

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

## Acknowledgements

C.L.M. acknowledges research support from Lyell Immunopharma, the Virginia and D.K. Ludwig Fund for Cancer Research, and the St. Baldrick's Foundation to the EPICC (Empowering Immunotherapies for Children's Cancers Cancer) Team. C.L.M is a member of the Parker Institute for Cancer Immunotherapy, which supports the Stanford University Cancer Immunotherapy Program. J.G. acknowledges the Blavatnik Family Foundation for financial support. K.J.S. acknowledges research support from the National Instutes of Health (R01DK125260, P30DK116074). All authors acknowledge Metafora Biosystems for generously providing the GLUT1 ligand for this study.

## Author contributions

Conceptualization: D.K., C.L.M. Methodology: J.G., D.K., E.S., M.M., L.S., B.M., K.C., S.A.F., M.K., K.D. Investigation: J.G., D.K., Carley F., M.B., P.X., J.H., S.D., Chris F., A.M., J.L., K.S. Visualization: J.G., D.K., Y.C., E.S. Writing, drafting: J.G., E.S., D.K., C.L.M. Writing, review, and editing: J.G., Y.C., E.S., D.K., C.L.M.

## Competing interests

C.L.M. is a cofounder of and holds equity in Link Cell Therapies, CARGO therapeutics and GBM NewCo. E.S holds equity in Lyell Immunopharma and consults for Lepton Pharmaceuticals and Galaria. S.A.F. serves on the Scientific Advisory Boards for Alaunos Therapeutics and Fresh Wind Biotech and has equity interest in both; S.A.F. receives research funding from CARGO and Tune Therapeutics. C.L.M. receives research funding from Lyell, Tune and consults for CARGO, Link, GBMNewCo, Immatics, and Ensoma. K.L.D is a St. Baldrick's Foundation Scholar and acknowledges support from Hyundai Hope on Wheels and Oxnard Foundation (K.L.D.). K.C. is currently employed by AstraZeneca. The remaining authors declare no competing interests.
