## [Peer Review File · Nature Communications]

GLUT1 overexpression in CAR-T cells induces metabolic reprogramming and enhances potencyREVIEWER COMMENTS

Reviewer #1 (Remarks to the Author):

The authors established GLUT1-overexpressing CAR-T cells (GLUT1 OE) and tried to characterize the metabolism, redox pathway, urea cycle, and anti-tumor effect of the GLUT1 OE, compared to those of control CAR T cells. Although various pathway modifications by GLUT1 overexpression have been identified, attempts to explain them in an integrated manner have not always been successful.

Major points

1. The fact that GLUT1 overexpression enhances mitochondrial oxidative phosphorylation is interesting, but perhaps less surprising given that it also promotes aerobic glycolysis.
2. The relationship between increased urea cycle and increased mTORC1 activity is not well understood. Metabolome analysis by mass spectrometry revealed apparently reduced arginine level in both CD19-Glut-1 and HA-Glut-1 CAR-T cells compared to that of control cells (Fig.4), while urea cycle is highly enriched. The authors believe that arginine is responsible for the increased activation of mTORC1, but the experimental evidence shows that arginine clearly decreases, making the argument unreasonable.
3. The pathway by which GSH is produced from γ -Glu-Cys requires both cysteine and glutamate (Supplemental Figure 2D). Glutamate in this case is derived from glutamine, so glutaminolysis is important to eliminate ROS (JCI 131: e140100, 2021). In addition, aKG, a product of glutaminolysis, is known to activate GPx to eliminate ROS, which results in promotion of cell proliferation (Cancer Cell 27, 257, 2015). Also, related to 1, aKG produced from glutaminolysis is known to activate mTORC1, promoting cell proliferation (Mol Cell,47,349,2012, J Immunother Cancer 9:e002954, 2021). Glutaminolysis is thus an essential pathway for interpreting the overall picture of the observed phenomenon, but this paper does not analyze or discuss this issue.
4. The authors should monitor mitochondrial ROS level by GLUT1 overexpression.
5. The authors should examine whether an inhibitor to G6PDH (blocking pentose phosphate pathway) suppresses IL-2 production of CD19-Glut1 cells in Figure 3

Reviewer #2 (Remarks to the Author):

In this study, Guerrero et al. investigate the impact of ectopic glucose transporter 1 overexpression (GLUT1-OE) on chimeric antigen receptor (CAR) expressing T cells both in vitro and in a xenograft tumor model. Mechanistically, the authors establish a connection between GLUT1-mediated glucose metabolism and the enhanced metabolic fitness of CAR T cells, leading to improved functionality. Notably, GLUT1-OE not only elevated glycolytic metabolism, as anticipated, but also increased mitochondrial respiration, glutathione generation, as well as urea and arginine metabolism. These metabolic adaptations confer increased resistance of GLUT1-OE CAR T cells to reactive oxygen species (ROS)-induced dysfunction, while enhancing their "stemness" potential. Finally, the authors demonstrate that GLUT1-OE enhances the antitumor efficacy of one CAR construct in a murine model of B-ALL (NALM6), underscoring the potential of GLUT1-OE in CAR T cells as a viable approach to enhancing tumor immunotherapy.

Overall, this is a well-designed study with interesting data that highlight the potential of metabolically engineered CAR T cells by GLUT1-OE. However, my enthusiasm for this study is a bit tempered because several studies have already demonstrated that overexpression of glucose transporters boost the functionality, stemness and longevity of T cells. For example, Jacobs et al. demonstrated that GLUT1-OE supports the metabolic fitness of T cell, effector function and promotes a memory-like phenotype in older mice (PMID: 18354169). Siska et al. showed that GLUT1-OE increases the production of IL-2, IFN γ in T cells during antitumor immunity (PMID: 27511728). Ectopic expression of GLUT3 in CD4 T cells promotes the effector function of Th17 cells (PMID: 35316657). GLUT3-OE also increases glucose uptake, glycolysis, glycogen and fatty acid storage, and is associated with increased mitochondrial fitness of CD8 T cells. Ectopic expression of GLUT3 also reduced ROS levels in CD8 T cells, providing a better resistance to redox stress and significantly increased antitumor immunity of CTLs in vivo (PMID: 36203587). Despite

not being CAR T cells, the findings of these studies are similar in nature as the findings provided in the current manuscript.

Although the data provided in this manuscript is clear and mostly convincing, there are some technical and conceptual shortcomings that should be addressed. I hope the following comments are helpful to improve the manuscript.

Major Points:

- 1) The authors use 2-NBDG labelling as a (indirect) measure for glucose uptake (Fig. 1B). However, 2-NBDG is not an accurate method to investigate glucose uptake as competitive substrates or GLUT inhibitors cannot halt 2-NBDG uptake, suggesting that the glucose analogue is taken up by other mechanisms (PMID: 32879737). Thus, the glucose uptake capacity should be examined using [3H]-2DG in the two GLUT1-OE CAR T cell models.
- 2) The Seahorse extracellular flux analyses to determine glycolytic activity (Fig. 1C) should be performed using a "glycolytic stress test" protocol (addition of glucose, oligomycin and 2-DG). Also, the additions of compounds at the different timepoints in Fig. 1C and Fig. 1E should be indicated in the figure or legend. Although these analyses have been performed with only one or two donors, the quantification and the statistical analyses of these data is based on technical replicates and/or the repeat measurements of the different condition. Since this is a key experiment, I think this should be repeated with more independent donors.
- 3) The presentation of the bulk RNA-seq analyses in Fig. 2A appears confusing. While it is apparent that the differences in gene expression are more pronounced in the CD19 CAR (blue) compared to the HA construct (red), the current representation suggests that GLUT1-OE has divergent effects on both CARs. Moreover, the Venn diagrams in Suppl. Fig. 2F indicate minimal overlap in gene expression following GLUT1-OE in the two CAR models. Numerous genes are up- and downregulated in the GLUT1-OE HA CAR T cells, yet these show little overlap with the gene expression patterns observed in the CD19 CAR T cells.
- 4) The metabolome experiments conducted in this study are intriguing, revealing unexpected impacts on various metabolic pathways after GLUT1-OE. However, it is surprising that only a limited number of glycolytic or TCA metabolites showed significant changes in these assays. To further elucidate the direct effects of heightened glucose uptake in GLUT1-OE CAR T cells on metabolic pathways, conducting [13C] glucose tracing LC/MS experiments would be appropriate. Performing analyses of intracellular [13C]-labelled metabolites following short (5-10 min), medium (2-6 h), and long (12-24 h) labeling periods will provide insights into the alterations within glycolysis, the TCA cycle, and the PPP. This approach would offer a clearer understanding of how GLUT1-OE modulates these key metabolic pathways.
- 5) How does elevated glucose uptake and glycolysis in GLUT1-OE T cells drive the far-reaching and complex genetic changes in CAR T cells? While altered glutathione biosynthesis and cellular ROS-detoxification may contribute significantly to the improved fitness of GLUT1-OE CAR T cells, it is important to ascertain whether these effects solely drive the functional improvements or if alterations in mitochondrial, urea, and arginine metabolism are equally involved. The present version of the study predominantly presents correlational findings rather than mechanistic insights.
- 6) The differences in gene expression and metabolism after GLUT1-OE appear to be more pronounced in the CD19 compared to the HA CAR construct. However, only HA CAR T cells were used for the NALM6 in vivo experiments. It would be good to also perform the comparison between the two CAR constructs in vivo.
- 7) Somewhat related to my previous comment, true challenge for (CAR) T cells lies in their utilization of glucose (and other metabolites) within solid tumors. It would greatly benefit the study to incorporate a model involving syngeneic solid cancer alongside the NALM6 xenograft model, preferably within an immunocompetent mouse model.
- 8) Previous studies have correlated heightened glucose consumption and glycolysis in T cells with

their terminal differentiation and exhaustion (PMID: 27452473, 37891230). Furthermore, restricting glucose uptake and utilization during (CAR) T cell production for immunotherapy has shown improvement in the quality and functionality of these cells by preventing functional exhaustion (PMID: 34233154, 32747793, 24091329, 25432172, 35032108, 37891230). While acknowledging that GLUT1-OE not only amplifies glycolysis but also enhances mitochondrial fitness and redox regulation in CAR T cells, it would be valuable to address these seemingly conflicting findings in the discussion.

9) The fact that GLUT1-OE may induce a memory-like phenotype in CAR T cells is intriguing but not very well explored. Does the RNA-sequencing data suggest a memory-shifted gene expression signature? The differences in TCF-1 and CD62L are also quite variable and not consistently significant. Relying solely on the modest differences in CD62L and TCF-1 expression in an in vitro culture to support the assertion that GLUT1-OE augments CAR T cell fitness and longevity via "stemness programming" (as stated in the abstract) may not be fully substantiated. Therefore, a more comprehensive examination of these markers, both in vitro and in vivo post-tumor antigen challenge, would provide a stronger basis for these claims.

10) The authors show that GLUT1-OE per se do not cause (spontaneous) exhaustion (Fig. 5D), but the conditions used in these experiments are also not expected to provoke T cell dysfunction. To corroborate the notion that GLUT1-OE CAR T cells may be more resistant to functional exhaustion, I suggest to chronically stimulate the CAR T cells with tumor antigens or antibodies in vitro or to analyze the GLUT1-OE CAR T cells after re-isolation from tumor-bearing mice. Both conditions should induce the upregulation of exhaustion markers and cause functional impairment (i.e. defects in proliferation, cytokine expression, killing capacity, apoptosis). I think these settings are more appropriate to evaluate the real potential and function of GLUT1-enhanced CAR T cells.

Finally, the strength of this study lies in its comparison of GLUT1-OE effects in two distinct CAR T cell models, acknowledging the influence of CAR design on heightened glucose consumption. However, some analyses are exclusively performed with only one construct, evident in Fig. 2B, C compared to Fig. 2D. Conducting parallel examinations of GLUT1-OE in both CAR models throughout the study would provide a more comprehensive assessment and comparison of GLUT1's effect on CAR T cells and significantly enhance the robustness of the study.

Minor Points:

- Fig. 3B: The labels on the histogram plots are difficult to read.
- Line: 149: Do you mean "transcriptomic" instead of "proteomic changes"?
- Fig. 3D,E and Suppl. Fig. 3C,D: Some of these effects are very minor. It would be nice to also present the original FACS data along with its quantification. Are there any differences between control and GLUT1-OE CAR T cells viability after H₂O₂ pre-conditioning?
- The labels of the pathways in Fig. 4A are barely readable. Also, the axis labels in Fig. 4C are too small.
- Does GLUT1-OE in CAR T cells also improve their proliferation and killing capacity and/or their resistance to apoptosis?

Reviewer #3 (Remarks to the Author):

The paper of Guerrero and colleagues describes the effect of co-overexpressing the glucose transporter GLUT1 with therapeutic CAR constructs. The main hypothesis of the authors is that rewiring the metabolism of therapeutic T cells would improve the CAR T cell capacity to operate in the challenging tumour microenvironment. GLUT1 is a known glucose transporter and its overexpression should provide a competitive advantage to capture glucose which will fuel T cell

effector functions. The authors claim that their hypothesis was correct since their CAR T cells appear improved in their metabolic capacity, which provides them improved functionality such as cytokine release and in vivo tumour control. In addition, the T cells did not appear to be more exhausted, and their activity was long lasting.

The topic of the paper is of interest and the modification of T cell to improve their function represents a clear unmet need. The fact that the "simple" overexpression of a glucose transporter provides the capacity to enhance CAR T cells is attractive. However, the reviewer is not convinced that the presented data are sufficient to lead to this conclusion. There are clear concerns about the interpretation of some data and there is a strong need for additional experiments.

General concerns:

- The authors show that glucose deprivation affects T cell expansion. Wouldn't it be more relevant to study CAR T cell lacking GLUT1 (by knocking it out)? This is even more relevant because the authors never show any situation where GLUT1 is completely absent. The fact that the tonic GD2CAR induces GLUT1 expression, but still exhausts the cells is not easy to assimilate in this context: if GLUT1 overexpression is sufficient to provide improved fitness, then why is GLUT1-induced expression (by tonic CAR) not able to do the same? Knockout experiments should be provided
- Although the in vivo experiment seems to support the main hypothesis, why did the authors use an artificial system (Nalm-6GD2) to show their point? Why not perform the experiment with the non-tonic CAR (CD19) with which the GLUT1 overexpression (and effect) is more obvious in vitro? The authors should run a "classical" in vivo experiment (showing tumour radiance and Kaplan Meier) comparing CD19CAR +/- GLUT1 and they should also include a mock control +/- GLUT1.
- Presence of GLUT1 should provide a competitive metabolic advantage of the CAR T cells over the tumour cells. This could be showed in vitro by co-culture assay. If this experiment reveals too complicated to run, the authors should at least discuss this point.

Major concerns:

- Statistics: although the statistical tests used are most of the time depicted, it is not clear why the authors sometimes use SEM and sometimes SD for error bars; normally only SD should be shown or the use of SEM should be clearly explained if kept. Specifically:
 - o F1B: stats between CD19 and CD19-Glut1 in both graphs show very small p values, however the distribution of the points does not seem to differ, can the authors check their calculations? The error bars (SEM which should be SD) are not visible.
 - o F1D: no stats but p values are shown.
 - o F4E: This quantification is not clear; the authors should show base-line or a 0% ASS1. If not possible, they should quantify the MFI. N=2 is probably not enough to run a t-test.
 - o F5B: "one representative donor of n=2-5" to plot a graph is not adequate, because the stats are run on experimental replicates rather than on experimental repeats.
- F5A-B: the functional assays are lacking important controls; The CARs minus GLUT1 show low cytokine release, but what is the baseline? The authors should add mock Tc from the same donors +/- GLUT1 and probably use additional target cells for CD19CAR. As mentioned before, it is important to show that GLUT1 alone is not affecting mock T cells. Here again, the inclusion of a knockout group would support the authors' hypothesis. The authors should also include some in vitro killing assays.
- F5C-F: Although the effect of HA-Glut1 over HA is remarkable, here again mock groups should be compared. F5E: no comments/explanations were provided concerning the HA+Glut1- population in the HA-Glut1 group. After the second re-challenge, where is animal 3 from GD2- group? This re-challenge provides important data, but this is one experiment, the reviewer wonders if this is enough to conclude about improved persistence.
- F2 and related text: the reviewer has no expertise, but thinks that the text should be made more accessible to readership.

Minor concerns:

- Graphical abstract: what does the yellow octagon represent in the cleft of the CAR construct?
- L.89: 2A is a ribosome skipping sequence, not "cleavage"

Response to Reviewer Comments

We would like to thank the reviewers for their thoughtful critiques and suggestions. We have used this guidance to significantly improve our manuscript. Our revised manuscript includes a substantial number of new experiments to strengthen our central finding, which is that GLUT1 overexpression in CAR T cells increases glucose consumption, glycolysis, glycolytic reserve and oxidative phosphorylation and this is associated with enhanced CAR T cell potency as measured by cytokine production and cytotoxicity *in vitro*, and tumor control and persistence *in vivo*. GLUT1 OE induces broad metabolic reprogramming and increased glutathione-mediated resistance to reactive oxygen species but does not predispose T cells to exhaustion. We have further added new data in the revised manuscript demonstrating increased Th₁₇ differentiation and increased inosine generation with GLUT1OE. New data incorporated into the revised manuscript is summarized below:

- Seahorse Glycolytic Stress Test confirming that GLUT1OE increases glycolytic reserve in CAR-T cells
- Phenotypic and transcriptomic analyses and antigen induced cytokine secretion demonstrating that GLUT1OE drives differentiation into Th17 phenotype
- Demonstration that GLUT1OE decreases mitochondrial ROS accumulation in CD19 CAR after stimulation.
- Demonstration that GLUT1OE increases glutaminolysis in CD19.28z.-CAR T cells, as evidenced by reduction of L-glutamine and increased mRNA levels of GLUD1 and other mitochondrial aminotransferases that catalyze the deamidation of glutamate (GOT2 and GPT2).
- Demonstration that GLUT1OE induced enhanced cytokine production does not require pentose phosphate pathway activity
- New analyses of bulk RNAseq dataset focusing on GLUT1OE-mediated changes common to both CARs, which shows downregulation of the *NK-like dysfunction* signature associated with T cell exhaustion (Good et al, Cell 2021). This finding was also validated experimentally in serial restimulation assays, which showed that GLUT1OE does not accelerate the onset of exhaustion but rather delays expression of exhaustion markers on CAR-T cells and induces an increased proportion of cell with a central memory phenotype.
- [U13C] glucose tracing demonstrate that increased intracellular glucose is distributed through conventional pathways, with the exception of a significant accumulation of inosine, which we have previously shown aids in T cell memory formation and alleviates T cell exhaustion (Klysz, Cancer Cell 2024).
- Additional models of solid and liquid tumors demonstrating that GLUT1OE increases persistence of CAR-T cells, and drives a memory phenotype.
- New data demonstrating that GLUT1OE increases GPC2-CAR-T activity against solid tumor lines with varying antigen densities.

Reviewer #1 (Remarks to the Author):

The authors established GLUT1-overexpressing CAR-T cells (GLUT1 OE) and tried to characterize the metabolism, redox pathway, urea cycle, and anti-tumor effect of the GLUT1 OE, compared to those of control CAR T cells. Although various pathway modifications by GLUT1 overexpression have been identified, attempts to explain them in an integrated manner have not always been successful.

Comment 1.1: *The fact that GLUT1 overexpression enhances mitochondrial oxidative phosphorylation is interesting, but perhaps less surprising given that it also promotes aerobic glycolysis.*

Response 1.1: We agree with the reviewer that this result is potentially not unexpected. The evidence for increased oxidative phosphorylation is strong and likely important for understanding the overall biology of GLUT1OE. Indeed, reviewer 2 suggested (**Comment 2.8**) that we further highlight this finding as a potential basis to explain the lack of enhanced exhaustion observed with GLUT1OE, which we now do in the discussion of the revised manuscript.

Comment 1.2: *The relationship between increased urea cycle and increased mTORC1 activity is not well understood. Metabolome analysis by mass spectrometry revealed apparently reduced arginine levels in both CD19-Glut-1 and HA-Glut-1 CAR-T cells compared to that of control cells (Fig.4), while urea cycle is highly enriched. The authors believe that arginine is responsible for the increased activation of mTORC1, but the experimental evidence shows that arginine clearly decreases, making the argument unreasonable.*

Response 1.2: We agree with the reviewer that we have not proven a causal relationship between increased utilization of arginine and increased activation of mTORC1 and thus we have removed this claim.

Comment 1.3: The pathway by which GSH is produced from γ -Glu-Cys requires both cysteine and glutamate (Supplemental Figure 2D). Glutamate in this case is derived from glutamine, so glutaminolysis is important to eliminate ROS (JCI 131: e140100, 2021). In addition, aKG, a product of glutaminolysis, is known to activate GPx to eliminate ROS, which results in promotion of cell proliferation (Cancer Cell 27, 257, 2015). Also, related to 1, aKG produced from glutaminolysis is known to activate mTORC1, promoting cell proliferation (Mol Cell, 47, 349, 2012, J Immunother Cancer

9:e002954, 2021). Glutaminolysis is thus an essential pathway for interpreting the overall picture of the observed phenomenon, but this paper does not analyze or discuss this issue.

Response 1.3: We thank the reviewer for raising this important issue. To directly address if glutaminolysis is increased due to GLUT1OE in the revised manuscript, we first analyzed the expression of key enzymes of the glutamine/glutamate metabolism in our RNAseq dataset. We observed that all 3 mitochondrial aminotransferases GOT2, GLUD1 and GPT2 are significantly upregulated in CD19-GLUT1OE CAR-T cells 4 hours post stimulation and also at baseline (**Figure R1.3.1**). Secondly, we now provide evidence of increased glutaminolysis based upon our mass spectroscopy data, which shows significantly less L-Glutamine in CD19-GLUT1OE plays (**Figure R1.3.2**). These data are consistent with a model wherein GLUT1OE increases glutaminolysis and antioxidant production. This data been added to the revised manuscript (**Supplementary Figure 4A-B**) and we further address this issue in the discussion.

Comment 1.4: The authors should monitor mitochondrial ROS level by GLUT1 overexpression.

Response 1.4: Thank you for this suggestion. In the revised manuscript, we have now measured mitochondrial ROS by intracellular flow cytometry (**New Figure R1.4 / 3E**) and saw no difference in ROS in CD19-CAR +/- GLUT1OE at baseline, however upon idiotype stimulation, GLUT1OE decreases ROS accumulation ($n=4$ donors). In the tonic signaling HA-CAR, GLUT1OE decreased ROS accumulation at baseline but had a less striking effect upon CAR stimulation. This experiment is independent from the data shown in R1.4 and reflects different donors. We conclude that GLUT1OE reduces mitochondrial ROS in activated T cells further supporting evidence for reduced ROS with GLUT1-OE.

Comment 1.5: The authors should examine whether an inhibitor to G6PDH (blocking pentose phosphate pathway) suppresses IL-2 production of CD19-Glut1 cells in Figure 3.

Response 1.5: Following the reviewer's suggestion, we cocultured CD19-CAR \pm GLUT1OE with Nalm6 leukemia \pm 10uM 6-aminonicotinamide (6-AN), a pentose phosphate pathway blocker, then assessed cytokine secretion 24h later by ELISA. We observed that GLUT1OE increased cytokine production by CD19 CAR-T cells with or without 6-AN (**Figure R1.5**). Although there was a trend of decreased IL-2 for both CD19 and GLUT1OE

CAR in the presence of 10 uM 6-AN, however the data collected did not meet statistical significance. We now include this data in the revised manuscript in **Figure 3H** and added language to state that the pentose phosphate pathway was not required for the enhanced cytokine production observed in GLUT1OE T cells.

Reviewer #2 (Remarks to the Author):

In this study, Guerrero et al. investigate the impact of ectopic glucose transporter 1 overexpression (GLUT1-OE) on chimeric antigen receptor (CAR) expressing T cells both *in vitro* and in a xenograft tumor model. Mechanistically, the authors establish a connection between GLUT1-mediated glucose metabolism and the enhanced metabolic fitness of CAR T cells, leading to improved functionality. Notably, GLUT1-OE not only elevated glycolytic metabolism, as anticipated, but also increased mitochondrial respiration, glutathione generation, as well as urea and arginine metabolism. These metabolic adaptations confer increased resistance of GLUT1-OE CAR T cells to reactive oxygen species (ROS)-induced dysfunction, while enhancing their "stemness" potential. Finally, the authors demonstrate that GLUT1-OE enhances the antitumor efficacy of one CAR construct in a murine model of B-ALL (NALM6), underscoring the potential of GLUT1-OE in CAR T cells as a viable approach to enhancing tumor immunotherapy. Overall, this is a well-designed study with interesting data that highlight the potential of metabolically engineered CAR T cells by GLUT1-OE. However, my enthusiasm for this study is a bit tempered because several studies have already demonstrated that overexpression of glucose transporters boost the functionality, stemness and longevity of T cells. For example, Jacobs et al. demonstrated that GLUT1-OE supports the metabolic fitness of T cell, effector function and promotes a memory-like phenotype in older mice (PMID: 18354169). Siska et al. showed that GLUT1-OE increases the production of IL-2, IFN γ in T cells during antitumor immunity (PMID: 27511728). Ectopic expression of GLUT3 in CD4 T cells promotes the effector function of Th17 cells (PMID: 35316657). GLUT3-OE also increases glucose uptake, glycolysis, glycogen and fatty acid storage, and is associated with increased mitochondrial fitness of CD8 T cells. Ectopic expression of GLUT3 also reduced ROS levels in CD8 T cells, providing a better resistance to redox stress and significantly increased antitumor immunity of CTLs *in vivo* (PMID: 36203587). Despite not being CAR T cells, the findings of these studies are similar in nature as the findings provided in the current manuscript. Although the data provided in this manuscript is clear and mostly convincing, there are some technical and conceptual shortcomings that should be addressed. I hope the following comments are helpful to improve the manuscript.

RESPONSE: We kindly thank the reviewer for highlighting the merits of this manuscript. As the reviewer points out, although previous works from other groups described the effect of GLUT1-OE in T cells, our work addresses GLUT1OE in the context of human CAR-T cells. Between the time of original submission and submission of this revision, a report describing effects of GLUT1 overexpression in human CAR-T cells was published (PMID: 38720457). The published report as well as our demonstrate that GLUT1OE increases OXPHOS, antitumor efficacy, and promotion of memory formation. Our report provides significant additional insights by demonstrating the variable effects in non-tonically signaling vs. tonically signaling CAR T cells, providing evidence that GLUT1OE does not accelerate T cell exhaustion, providing new insights into the metabolic reprogramming induced by GLUT1OE, including demonstrating reduced reactive oxygen species, demonstrating enhanced glycolytic reserve, and by glucose tracing we provide the novel finding that GLUT1OE increases inosine generation in T cells. Further, we include new data in the revision demonstrating that GLUT1OE drives Th17 differentiation, which has not been reported previously.

Comment 2.1: The authors use 2-NBDG labelling as a (indirect) measure for glucose uptake (Fig. 1B). However, 2-NBDG is not an accurate method to investigate glucose uptake as competitive substrates or GLUT inhibitors cannot halt 2-NBDG uptake, suggesting that the glucose analogue is taken up by other mechanisms (PMID: 32879737). Thus, the glucose uptake capacity should be examined using [3H]-2DG in the two GLUT1-OE CAR T cell models.

Response 2.1: As suggested by the reviewer, we measured glucose internalization \pm GLUT1OE in bulk (mixed CD4 and CD8) MOCK and CAR-T cells from 4 healthy donors +/- stimulation for 24h using [3H]-2DG. The results aligned with what we have previously observed with the 2-NBDG labeling (Fig R2.1): GLUT1OE increased glucose internalization in Mock-T cells and CD19-CAR T cells in the unstimulated state, and increased glucose internalization in CD19-CAR cells following stimulation and tonic signaling drives greater glucose uptake in HA

CAR T cells than in MOCK and CD19. The only discrepancy between the two assays, was the absence of significant increase in uptake of $[3H]-2DG$ by HA-CAR T cells \pm GLUT1OE, which we attribute to the increased GLUT1 at baseline due to tonic signaling, and thus GLUT1OE has lesser effects (**Figure 1A**). This new data is **Figure S1B** in the updated manuscript.

Comment 2.2: The Seahorse extracellular flux analyses to determine glycolytic activity (Fig. 1C) should be performed using a “glycolytic stress test” protocol (addition of glucose, oligomycin and 2-DG). Also, the additions of compounds at the different timepoints in Fig. 1C and Fig. 1E should be indicated in the figure or legend. Although these analyses have been performed with only one or two donors, the quantification and the statistical analyses of these data is based on technical replicates and/or the repeat measurements of the different condition. Since this is a key experiment, I think this should be repeated with more independent donors.

Response 2.2: We thank the reviewer for these suggestions. We’ve now performed Mitostress Tests with additional donors, with revised **Figure 1D** showing pooled data of n=5-6 donors. The results from additional experiments strengthened our initial findings of increased basal, maximal and spare respiratory oxygen consumption rates in Glut1OE CAR-T cells (**Fig. 1E**) and increased basal ECAR in tonic signaling HA-CAR-T cells but not CD19-CAR-T cells (**Fig. 1C**). We also performed Seahorse Glycolytic stress test as suggested by the reviewer, and found that glycolytic reserve is increased in Glut1OE CAR-T cells, although, more striking for CD19- as compared to HA-CAR-T cells (**Fig2.2**). Together, the data demonstrate that Glut1OE cells exhibit increased mitochondrial activity, glycolytic reserve and glycolytic flux upon activation (or tonic signaling).

Additionally, we have updated the figures to indicate addition of inhibitors throughout Seahorse assays as requested.

Comment 2.3: The presentation of the bulk RNA-seq

analyses in Fig. 2A appears confusing. While it is apparent that the differences in gene expression are more pronounced in the CD19 CAR (blue) compared to the HA construct (red), the current representation suggests that GLUT1-OE has divergent effects on both CARs. Moreover, the Venn diagrams in Suppl. Fig. 2F indicate minimal overlap in gene expression following GLUT1-OE in the two CAR models. Numerous genes are up- and downregulated in the GLUT1-OE HA CAR T cells, yet these show little overlap with the gene expression patterns observed in the CD19 CAR T cells.

Response 2.3: We appreciate the reviewer’s comment and apologize for the lack of clarity. With regard to the minimal overlap in gene expression, this aligns with extensive previous data from our group showing the dramatic changes in phenotype, function, transcriptome and epigenome between non-tonically signaling CARs (e.g.

CD19-CAR) and those that undergo tonic signaling (e.g. HA-CARs) (Lynn, Nature, 2019; Weber Science 2021). Thus, these are two prototypes for “normal” vs “exhausted” CAR-T cells, respectively, and represent extremes on the spectrum of CAR activities at baseline, before GLUT1OE. The divergence in their baseline transcriptome profiles is consistent with previous work and explains the divergent responses to GLUT1OE. In addition however, we agreed with the reviewer that

additional insight could be gleaned from the RNA-Seq data. Hence, in the updated manuscript, we have replaced the Venn diagrams with UpSet plots and reanalyzed the RNAseq data focusing on the similarities between CD19 and HA CAR-T cells upon GLUT1OE. We identified 844 differentially expressed genes (406 upregulated and 438 downregulated, highlighted in **Fig. R2.3 left panel**) shared by CD19-CAR and HA-CAR at baseline. Gene set enrichment analysis of these 800 genes showed that GLUT1OE downregulates the gene signature associated with T cell exhaustion (Good et al, Cell 2021) in both CARs (**Fig. R2.3 right panels**). This reformatted data is now **Figure 2A-B** in the updated manuscript. Given the concern that increased glucose uptake might drive T cell exhaustion, these results are of great interest.

Comment 2.4: *The metabolome experiments conducted in this study are intriguing, revealing unexpected impacts on various metabolic pathways after GLUT1-OE. However, it is surprising that only a limited number of glycolytic or TCA metabolites showed significant changes in these assays. To further elucidate the direct effects of heightened glucose uptake in GLUT1-OE CAR T cells on metabolic pathways, conducting [13C] glucose tracing LC/MS experiments would be appropriate. Performing analyses of intracellular [13C]-labelled metabolites following short (5-10 min), medium (2-6 h), and long (12-24 h) labeling periods will provide insights into the alterations within glycolysis, the TCA cycle, and the PPP. This approach would offer a clearer understanding of how GLUT1-OE modulates these key metabolic pathways.*

Response 2.4: We agreed with the reviewer that global LC/MS analyses would shed light on the effect that GLUT1OE has on the balance between glycolysis, the TCA cycle and the PPP. As we did not have these assays available in our laboratory, we worked with General Metabolics (Cambridge, MA) and performed U-¹³C Glucose tracing in unstimulated and stimulated (15min and 4h) CD19 +/- GLUT1OE in CAR-Ts from 3 donors (detailed procedures can be found in the methods section of the manuscript). Unfortunately, samples of CAR-T cells stimulated in the presence of glucose isotopes for 15min didn't yield measurable amounts of any labeled metabolite. However, the 4h-stimulation condition yielded interesting data, demonstrating that GLUT1OE promoted lactic acid formation, and glutamate derived from TCA activity. The biosynthesis of inosine, indirectly formed through the PPP, was also found in greater abundance in CD19.28ζ-GLUT1, and significantly so after stimulation. Our data are consistent with a model wherein the bulk of changes observed in GLUT1OE CAR-T

cells do not result from unrestrained upregulation of metabolic enzymes, but rather the flux of glucose-derived metabolites cycling through the PPP, TCA and urea cycles. (**Fig R2.4**). This data is included in the updated manuscript as **Figure 4D**. Additional insights using this comprehensive approach could be gleaned with additional experiments and conditions (e.g. longer timepoints, ± glucose starvation etc.) but due to costs and time involved, we believe that additional studies characterizing the effects of GLUT1OE using LC/MS are beyond the scope of this manuscript.

Comment 2.5: *How does elevated glucose uptake and glycolysis in GLUT1-OE T cells drive the far-reaching and complex genetic changes in CAR T cells? While altered glutathione biosynthesis and cellular ROS-detoxification may contribute significantly to the improved fitness of GLUT1-OE CAR T cells, it is important to ascertain whether these effects solely drive the functional improvements or if alterations in mitochondrial, urea, and arginine metabolism are equally involved. The present version of the study predominantly presents correlational findings rather than mechanistic insights.*

Response 2.5: We agree that additional mechanistic insights dissecting the downstream effect of GLUT1OE could be garnered through future work. However, this manuscript provides a comprehensive analysis of the effects of GLUT1-OE on CAR T cells, demonstrating increased glycolysis, oxidative phosphorylation, decreased mitochondrial reactive oxygen species, increased cytokine production and tumor control and in the revised manuscript, new data showing induction of Th17 differentiation. We further show significant differences in the biology based upon the activation state of the T cell. While additional work could be performed to attempt to

identify specific metabolic pathways responsible for each of these effects, we respectfully posit that such work is beyond the scope of this manuscript.

Comment 2.6: The differences in gene expression and metabolism after GLUT1-OE appear to be more pronounced in the CD19 compared to the HA CAR construct. However, only HA CAR T cells were used for the NALM6 *in vivo* experiments. It would be good to also perform the comparison between the two CAR constructs *in vivo*.

Response 2.6: Following the reviewer's suggestion, we performed *in vivo* stress tests, whereby a suboptimal doses of CD19-CAR T cells ($0.35e^6$) are infused in NSG mice bearing Nalm-6 leukemia ($1e^6$ cells delivered *iv*). BLI monitoring of tumor growth showed a modest but significant delay in tumor progression in mice injected with CD19-Glut1 CAR-T cells (Fig. R2.6 left panel).

Furthermore, analysis of spleens at endpoint showed 5-times more CAR-T cells in the GLUT1OE group than in the CD19-CAR only group. Also, we found negligible amounts of tumor cells in the spleens of GLUT1OE group, compared to ~40% Nalm6 in the spleens of CD19 CAR-T treated mice (Fig. R2.6 right panels). These data can now be found in **Figure 6A-C** of the revised manuscript.

Comment 2.7: Somewhat related to my previous comment, true challenge for (CAR) T cells lies in their utilization of glucose (and other metabolites) within solid tumors. It would greatly benefit the study to incorporate a model involving syngeneic solid cancer alongside the NALM6 xenograft model, preferably within an immunocompetent mouse model.

Response 2.7: While we agree that a syngeneic, immunocompetent mouse model would bring insight to our work and bolster our claims, the time and money required to create murine versions of our reagents are beyond the scope for this original manuscript, which comprehensively analyzes the effects of GLUT1OE in human CAR T cells.

When challenged *in vitro* with tumor lines expressing various antigen densities (~6.8K and ~34K molecules of GPC2), we observed increased cytotoxic activity in an Incucyte assay at multiple E:T ratios and increased cytokine secretion (n=3 donors) (**Fig R2.7A-C**). When tested *in vivo* against the low antigen expressing cell line, SMS-SAN, we observed similar tumor control, but increased peripheral persistence 18 days post challenge as well as increased % of T_{SCM} population 39 days post infusion (**Fig R2.7D-E**). **The new *in vitro* data is in Figure 5 E-H, and the *in vivo* in 6G-H.**

Comment 2.8: Previous studies have correlated heightened glucose consumption and glycolysis in T cells with their terminal differentiation and exhaustion (PMID: 27452473, 37891230). Furthermore, restricting glucose uptake and utilization during (CAR) T cell production for immunotherapy has shown improvement in the quality and functionality of these cells by preventing functional exhaustion (PMID: 34233154, 32747793, 24091329, 25432172, 35032108, 37891230). While acknowledging that GLUT1-OE not only amplifies glycolysis but also enhances mitochondrial fitness and redox regulation in CAR T cells, it would be valuable to address these seemingly conflicting findings in the discussion.

Response 2.8: We agree with the reviewer that the findings were not entirely predicted based upon data emanating from glucose restriction experiments. We have extensively tested if GLUT1OE CAR-T cells are more prone to exhaustion in vitro with serial stimulation assays and co-cultures at low E:T, and in vivo, utilizing stress tests with suboptimal doses, and we've consistently observed enhanced potency, persistence and delayed onset of exhaustion (new data in **Figures 5** and **6**) We have now added new text in the discussion section highlighting this interesting result that challenges current paradigms.

Comment 2.9: *The fact that GLUT1-OE may induce a memory-like phenotype in CAR T cells is intriguing but not very well explored. Does the RNA-sequencing data suggest a memory-shifted gene expression signature? The differences in TCF-1 and CD62L are also quite variable and not consistently significant. Relying solely on the modest differences in CD62L and TCF-1 expression in an in vitro culture to support the assertion that GLUT1-OE augments CAR T cell fitness and longevity via "stemness programming" (as stated in the abstract) may not be fully substantiated. Therefore, a more comprehensive examination of these markers, both in vitro and in vivo post-tumor antigen challenge, would provide a stronger basis for these claims.*

Response 2.9: In the revised version of our manuscript, we have included additional data confirming that a small but significant subset of GLUT1OE CAR-T cells differentiate towards memory phenotype. First, GSEA analysis of RNA-Seq from CD19-CAR revealed increased memory and effector transcriptomic programming with GLUT1-OE, especially in unstimulated cells (**Fig. R2.9.1**).

Second, analyses of the expression of CD62L, CCR7 and CD45RA expression on CD19-CAR +/- GLUT1OE after four consecutive challenges with CD19+ tumor cells showed an increase in Central Memory (CD62L+, CCR7+, CD45RA-) cell populations in both CD8+ and CD4+ CD19-GLUT1OE cells (n=3 donors) (**Fig. R2.9.2**). Parallel analysis was done in HA CAR-T cells after three consecutive stimulations with Nalm6-GD2, where we found enrichment in Central and Effector Memory populations, but only in the CD4+ compartment. These data are now **Figure 5D, S2B** in the updated manuscript. In addition, we have removed the claim of "stemness programming" from the manuscript given the concern raised by the reviewer.

Comment 2.10: *The authors show that GLUT1-OE per se does not cause (spontaneous) exhaustion (Fig. 5D), but the conditions used in these experiments are not expected to provoke T cell dysfunction. To corroborate the notion that GLUT1-OE CAR T cells may be more resistant to functional exhaustion, I suggest to chronically stimulate the CAR T cells with tumor antigens or antibodies or to analyze the GLUT1-OE CAR T cells after re-isolation from tumor-bearing mice. Both conditions should induce the upregulation of exhaustion markers and cause functional impairment (i.e. defects in proliferation, cytokine expression, killing capacity, apoptosis). I think these settings are more appropriate to evaluate the real potential and function of GLUT1-enhanced CAR T cells.*

Response 2.10: We followed the reviewer's suggestion and monitored sequential tumor killing kinetics of Nalm6-GD2 by CD19 and HA CAR-Ts (1:2 E:T) in an Incucyte (n=4 donors) (**Fig R2.10**). After each round of tumor clearance, we performed flow cytometry analysis for CD39 expression which, based on our experience, is the most reliable marker of exhaustion in conditions of antigen stimulation (Klysz et al, Cancer Cell 2024). We found that after 6 consecutive challenges, CD19-GLUT1OE performed just as well as CD19 T cells derived from four different donors. Although killing capacity was unaffected, our data demonstrates that GLUT1OE cells exhibit lower expression of CD39, especially in the CD4 compartment. GLUT1OE allowed for improved tumor clearance for HA after one stimulation, then performed similarly until both CARs failed to control tumor after 3 stimulations. CD39 presentation was high for HA independent of GLUT1OE, but we recorded significant decreases up to 2 stimulations in the CD8 compartment. These data are now **Figure 5C, S6C** in the updated manuscript.

Comment 2.11: Finally, the strength of this study lies in its comparison of GLUT1-OE effects in two distinct CAR T cell models, acknowledging the influence of CAR design on heightened glucose consumption. However, some analyses are exclusively performed with only one construct, evident in Fig. 2B, C compared to Fig. 2D. Conducting parallel examinations of GLUT1-OE in both CAR models throughout the study would provide a more

comprehensive assessment and comparison of GLUT1's effect on CAR T cells and significantly enhance the robustness of the study.

Response 2.11: Thank you for this comment. To fill in the gaps highlighted by the reviewer we have now included the following data: RNA analysis has been modified to show converging effects of GLUT1OE in both CD19 and HA (See Responses 2.3 and 2.5 and Figures 2 and S2). New Seahorse data includes both CARs and MOCK +/-Glut1 (Figures 1 and S1), and we have generated new in vivo data using CD19-CARs in Figure 6A. Unfortunately, HA CAR-T cells are too sensitive to REDOX modifications and the majority of REDOX associated functional data was generated only from the CD19 CAR (Fig 3).

Comment 2.12: Fig. 3B: The labels on the histogram plots are difficult to read.

Response 2.12: We increased the sizes of labels to make it easier to read.

Comment 2.13: Line: 149: Do you mean "transcriptomic" instead of "proteomic changes"?

Response 2.13: Thank you, we corrected this mistake.

Comment 2.14: Fig. 3D,E and Suppl. Fig. 3C,D: Some of these effects are very minor. It would be nice to also present the original FACS data along with its quantification. Are there any differences between control and GLUT1-OE CAR T cells viability after H₂O₂ pre-conditioning?

Response 2.14: We have included a representative flow histogram showing IL-2 staining in CD19 CD4+ T cells +/-GLUT1OE +/-H₂O₂ preconditioning (Figure 3F). To address the reviewer's question about viability, we reanalyzed the flow cytometry data and assessed the percent of viable cells for each of the four donors and each of the conditions (Fig R2.14) (Fig S4C). We found no differences in viability comparing CD19 and CD19-GLUT1OE for each condition. We did observe a dose dependent viability decrease as the concentration of H₂O₂ is titrated up, with a rescue in the conditions containing the catalase control.

Comment 2.15: The labels of the pathways in Fig. 4A are barely readable. Also, the axis labels in Fig. 4C are too small.

Response 2.15: We apologize for this formatting error. We have increased the size of labels.

Comment 2.16: Does GLUT1-OE in CAR T cells also improve their proliferation and killing capacity and/or their resistance to apoptosis?

Response 2.16: In terms of proliferation, we determined that GLUT1OE did not provide any advantage during the exponential growth phase of CAR-T cell manufacturing across four independent donors at standard glucose concentration (11.11 mM) (**Fig R2.16**). To determine whether GLUT1OE affected apoptotic pathways, we stained for cleaved PARP, a readout of apoptosis, 19 days post start of manufacturing (when cell growth is stable) following a 15 hour CAR-based activation across three independent donors and found no differences between CAR-T cells ± GLUT1OE. In terms of killing capacity, we found that GLUT1OE enhanced tumor control at low E:T ratios when challenged with neuroblastoma cell lines bearing different amounts of GPC2 antigen (**New data in Figure 5G-H**).

Reviewer #3 (Remarks to the Author): The paper of Guerrero and colleagues describes the effect of co-overexpressing the glucose transporter GLUT1 with therapeutic CAR constructs. The main hypothesis of the authors is that rewiring the metabolism of therapeutic T cells would improve the CAR T cell capacity to operate in the challenging tumour microenvironment. GLUT1 is a known glucose transporter and its overexpression should provide a competitive advantage to capture glucose which will fuel T cell effector functions. The authors claim that their hypothesis was correct since their CAR T cells appear improved in their metabolic capacity, which provides them improved functionality such as cytokine release and in vivo tumour control. In addition, the T cells did not appear to be more exhausted, and their activity was long lasting. The topic of the paper is of interest and the modification of T cell to improve their function represents a clear unmet need. The fact that the “simple” overexpression of a glucose transporter provides the capacity to enhance CAR T cells is attractive. However, the reviewer is not convinced that the presented data are sufficient to lead to this conclusion. There are clear concerns about the interpretation of some data and there is a strong need for additional experiments.

RESPONSE: We thank the reviewer for their constructive comments and interest in our manuscript. We have added new experiments and analysis to address the reviewer’s concerns, which have substantially strengthened the manuscript.

Comment 3.1: The authors show that glucose deprivation affects T cell expansion. Wouldn’t it be more relevant to study CAR T cell lacking GLUT1 (by knocking it out)? This is even more relevant because the authors never show any situation where GLUT1 is completely absent. The fact that the tonic GD2CAR induces GLUT1 expression, but still exhausts the cells is not easy to assimilate in this context: if GLUT1 overexpression is sufficient to provide improved fitness, then why is GLUT1-induced expression (by tonic CAR) not able to do the same? Knockout experiments should be provided.

Response 3.1: At the beginning of this project we shared the reviewer’s reasoning and, to assess the dependency of CAR-T cells on glucose, we knocked-down GLUT1 with CRISPR-Cas9. We observed no significant functional changes by knockdown of GLUT1 alone. However, we hypothesized that because GLUT1 is one among a family of fourteen SLC2 nutrient transporters, there would be redundant functionality when one, or more, are unavailable. To test this hypothesis, we generated GLUT1-KO MOCK and CAR-T cells and cultured them in medium containing 11mM, 5.5mM or no glucose, and observed upregulation of GLUT3 in the Glucose free condition (**Fig R3.1, for reviewers only**). Thus, this functional redundancy made us pivot and redesign our experiments using limiting amounts of glucose in the media instead, to determine CAR-T cell dependency.

To answer the reviewer's second question about why HA-CAR T cells still get exhausted despite upregulating GLUT1, the easiest explanation would be that the upregulation driven by tonic signaling is insufficient to satisfy their metabolic needs. On the other hand, T cell exhaustion is multifactorial, and is not likely driven by limiting glucose alone. Thus, it may not be surprising that increasing glucose uptake in the very exhausted HA-CAR model does not fully reverse the phenotype, although it does seem to delay its onset, as shown in **Fig. R2.10**.

Despite these complexities, the effects of GLUT1OE presented here provide clear evidence that modulating this axis induces significant changes in T cell biology and enhances CAR T cell function.

Comment 3.2: *Although the in vivo experiment seems to support the main hypothesis, why did the authors use an artificial system (Nalm-6GD2) to show their point? Why not perform the experiment with the non-tonic CAR (CD19) with which the GLUT1 overexpression (and effect) is more obvious in vitro? The authors should run a “classical” in vivo experiment (showing tumour*

radiance and Kaplan Meier) comparing CD19CAR +/- GLUT1 and they should also include a mock control +/- GLUT1.

Response 3.2: A similar point was brought up by reviewer #2 and to address it we performed in vivo stress tests, whereby a suboptimal doses of CD19-CAR T cells (0.35×10^6) are infused in NSG mice bearing Nalm-6 leukemia (1×10^6 cells delivered iv). BLI monitoring of tumor growth showed a modest but significant delay in tumor progression in mice injected with CD19-Glut1 CAR-T cells in both models (Fig. R2.6 left and center panels). Furthermore, analysis of spleens at end-point showed 5-times more CAR-T cells in the GLUT1OE group than in the CD19-CAR only group. Also, we found negligible amounts of tumor cells in the spleens of GLUT1OE group, compared to ~40% Nalm6 in the spleens of CD19 CAR-T treated mice (Fig. R2.6). These data can now be found in **Figure 6A-C**.

Comment 3.3: *Presence of GLUT1 should provide a competitive metabolic advantage of the CAR T cells over the tumour cells. This could be showed in vitro by co-culture assay. If this experiment reveals too complicated to run, the authors should at least discuss this point.*

Response 3.3: We agree with the reviewer that GLUT1OE could provide an advantage in highly nutrient competitive tumor microenvironment. Unfortunately, setting up a reliable co-culture model in vitro is challenging and likely would not mimic the findings in vivo and thus we have chosen to address this point, as suggested, in the discussion.

Comment 3.4: *Statistics: although the statistical tests used are most of the time depicted, it is not clear why the authors sometimes use SEM and sometimes SD for error bars; normally only SD should be shown or the use of SEM should be clearly explained if kept. Specifically: o F1B: stats between CD19 and CD19-Glut1 in both graphs show very small p values, however the distribution of the points does not seem to differ, can the authors check their calculations? The error bars (SEM which should be SD) are not visible. o F1D: no stats but p values are shown.*

Response 3.4: Thank you for these comments. We have used SEM in panels representing pooled data from independent experiments and donors and SD when representing technical replicates or a representative donor.

Comment 3.5: *F4E: This quantification is not clear; the authors should show base-line or a 0% ASS1. If not possible, they should quantify the MFI. N=2 is probably not enough to run a t-test.*

Response 3.5: To better represent the flow data, the ASS1 expression is now shown as MFI and includes an FMO control. This did not affect our main conclusion that ASS1 is increased in Glut1OE CAR-T cells.

Comment 3.6: *F5B: “one representative donor of n=2-5” to plot a graph is not adequate, because the stats are run on experimental replicates rather than on experimental repeats.*

Response 3.6: With the reviewer suggestion we changed this figure from “representative donor” to fold change normalized to the levels of cytokine secreted by control CAR-T cell from 6-7 donors. This allowed us to account for a great difference of cytokine secretion between the donors (**new Figure 5A and B**).

Comment 3.7: F5A-B: the functional assays are lacking important controls; The CARs minus GLUT1 show low cytokine release, but what is the baseline? The authors should add mock Tc from the same donors +/- GLUT1 and probably use additional target cells for CD19-CAR. As mentioned before, it is important to show that GLUT1 alone is not affecting mock T cells. Here again, the inclusion of a knockout group would support the authors' hypothesis. The authors should also include some in vitro killing assays.

Response 3.7: As mentioned earlier in response to reviewer #2 in **response 2.16**, overexpression of Glut1 did not induce any effect on T cell proliferation, hence did not promote hyperproliferation due to increased carbon intake. Moreover, Mock-Glut1 overexpressing T cells do not express more cytokines over control without stimulation (**Fig. R3.7**, for reviewer only) nor affect tumor killing in vivo (**Fig. R2.6 and updated Fig. 6A**).

Comment 3.8: - F5C-F: Although the effect of HA-Glut1 over HA is remarkable, here again mock groups should be compared. F5E: no comments/explanations were provided concerning the HA+Glut1- population in the HA-Glut1 group. After the second re-challenge, where is animal 3 from GD2- group? This re-challenge provides important data, but this is one experiment, the reviewer wonders if this is enough to conclude about improved persistence.

Response 3.8: We included the requested Mock and Mock-Glut1 T cells in the Nalm6 stress test model described previously (**Fig.R2.6 and new Fig. 6A-C**) and we did not observe any effect of Mock-Glut1 on the tumor growth. Although we have only one re-challenge experiment, superior

Nalm6-GD2 control by GLUT1OE HA-CAR-T cells were shown in two independent experiments (**Fig. R3.8**). These experiments were done by co-transduction of two retroviral vectors, which do produce HA+GLUT1-populations, but these populations are smaller in proportion when compared to double positive populations (**Figure S7D**). The missing mice in the rechallenge experiment was found dead. We added that comment to the figure legend for clarity. To further validate our findings about increased persistence, we analyzed blood from GPC2-CAR treated mice and observed that, upon similar tumor control, the GLUT1OE-GPC2 treated mice showed significantly more CAR-T cells 18 days post-infusion than the GPC2-CAR group. This difference persisted 34 days later. Similarly, we encounter ~5-fold more CAR-T cells in the spleens from mice treated with GLUT1OE CD19-CAR than in the CD19-CAR group at endpoint. These new data are now shown in **Figure 6**.

Comment 3.9: - F2 and related text: the reviewer has no expertise but thinks that the text should be made more accessible to readership.

Response 3.9: We made significant changes to the transcriptomic data analysis by emphasizing more of a commonality between the CARs overexpressing Glut1 as compared to differences which result from different spectrum of the compared CARs (**Figure 2A-B, also Response 2.3 and 2.5**). This significantly should improve accessibility and understanding of the data.

Comment 3.10: - Graphical abstract: what does the yellow octagon represent in the cleft of the CAR construct?

Response 3.10: The yellow octagon was meant to depict an antigen. It has been removed for clarity.

Comment 3.11: - L.89: 2A is a ribosome skipping sequence, not "cleavage"

Response 3.11: Thank you noting this error. We have updated the text.

REVIEWERS' COMMENTS

Reviewer #1 (Remarks to the Author):

The authors performed additional new experiments to answer my questions. Now the revised manuscript has convincingly addressed my concerns.

Reviewer #2 (Remarks to the Author):

Response to the authors

The authors have done an excellent job at dealing with my and the other reviewer's comments. The main finding that GLUT1 overexpression boost CAR T cell function by inducing far-reaching effects on metabolism, transcription and stemness, is intriguing and clinically relevant.

There are a few interesting mechanistic questions that remain, but I agree these would go beyond the scope of a clinically oriented manuscript. Thus, I highly recommend the revised manuscript for publication in Nature Communications.

Additional comment: Please double check the colour coding in Fig 1d, middle panel. I have the feeling that the colours are swapped between the HA and HA-GLUT1OE groups.

Reviewer #3 (Remarks to the Author):

The authors have performed a substantial revision of the first version of their article. The reviewer is satisfied with the additional work and the answers, but would like to address two points:

- Comment 3.4: the reviewer does not agree with the use of SEM in the context proposed by the authors. Although often use to describe dispersion of the data, SEM will only inform you about the accuracy of your mean. It is therefore highly recommended to replace the SEM by SD. It will in any case not modify the statistical tests and will provide an accurate visualization of the variation.
- During the revision of this article, a paper using the same strategy was published in Molecular Therapy (PMID: 38720457), it would be of interest to provide a few sentences about the similarities and differences between the two studies.

REVIEWERS' COMMENTS

We are extremely pleased that all 3 reviewers find the revised version of the manuscript acceptable for publication in Nature Communication. Below are the point-by-point responses to the 3 remaining comments brought up by reviewers #2 and #3.

Reviewer #2 (Remarks to the Author):

Additional comment: Please double check the colour coding in Fig 1d, middle panel. I have the feeling that the colours are swapped between the HA and HA-GLUT1OE groups.

Response: Thank you for pointing this possible error. We have double checked all panels in Figure 1D, and throughout the paper, for accuracy.

Reviewer #3 (Remarks to the Author):

Comment: The authors have performed a substantial revision of the first version of their article. The reviewer is satisfied with the additional work and the answers but would like to address two points:

- Comment 3.4: the reviewer does not agree with the use of SEM in the context proposed by the authors. Although often use to describe dispersion of the data, SEM will only inform you about the accuracy of your mean. It is therefore highly recommended to replace the SEM by SD. It will in any case not modify the statistical tests and will provide an accurate visualization of the variation.

Following reviewer's suggestion we have updated all our Seahorse data with error bars reflecting SD, rather than SEM. This change affect all Seahorse plots and pooled data panels (Figures 1 and 2 and Supp Figure 1). Furthermore, we now also represent individual human donors as single data points, with lines connecting conditions +/- GLUT1 as per the Nature Communications' formatting guidelines. Lastly, we have updated our glucose tracing data with error bars reflecting SD.

- During the revision of this article, a paper using the same strategy was published in Molecular Therapy (PMID: 38720457), it would be of interest to provide a few sentences about the similarities and differences between the two studies.

We have updated our discussion with a reference for Shi et al's work in line 352 emphasizing the shared finding of memory formation and enhanced antitumor activity in CAR-T cells with GLUT1OE.